# LOOKING BACKWARD: RETROSPECTIVE BACKWARD SYNTHESIS FOR GOAL-CONDITIONED GFLOWNETS

**Haoran He**[1], **Can Chang**[2], **Huazhe Xu**[2], **Ling Pan**[1†]
[1]Hong Kong University of Science and Technology     [2]Tsinghua University

## ABSTRACT

Generative Flow Networks (GFlowNets), a new family of probabilistic samplers, have demonstrated remarkable capabilities to generate diverse sets of high-reward candidates, in contrast to standard return maximization approaches (e.g., reinforcement learning) which often converge to a single optimal solution. Recent works have focused on developing goal-conditioned GFlowNets, which aim to train a single GFlowNet capable of achieving different outcomes as the task specifies. However, training such models is challenging due to extremely sparse rewards, particularly in high-dimensional problems. Moreover, previous methods suffer from the limited coverage of explored trajectories during training, which presents more pronounced challenges when only offline data is available. In this work, we propose a novel method called **R**etrospective **B**ackward **S**ynthesis (**RBS**) to address these critical problems. Specifically, RBS synthesizes new backward trajectories in goal-conditioned GFlowNets to enrich training trajectories with enhanced quality and diversity, thereby introducing copious learnable signals for effectively tackling the sparse reward problem. Extensive empirical results show that our method improves sample efficiency by a large margin and outperforms strong baselines on various standard evaluation benchmarks. Our codes are available at `https://github.com/tinnerhrhe/Goal-Conditioned-GFN`.

## 1 INTRODUCTION

Generative Flow Networks (GFlowNets; Bengio et al. (2021)) are a new class of probabilistic models designed for sampling compositional objects from high-dimensional unnormalized distributions. GFlowNets generate each sample independently and amortize the sampling cost, and therefore do not suffer from the mixing problem (Salakhutdinov, 2009; Bengio et al., 2013; 2021) in Markov Chain Monte Carlo (MCMC) (Metropolis et al., 1953; Hastings, 1970; Andrieu et al., 2003). Indeed, GFlowNets transform sampling into a sequential decision-making problem: the agent learns a stochastic policy for sampling proportionally to the rewards, wherein each sequence of actions yields a unique object. In this regard, GFlowNets resemble reinforcement learning (RL) (Tiapkin et al., 2024; Deleu et al., 2024b; He et al., 2024), although standard RL typically focuses on optimizing policies for a single reward-maximizing objective. Due to the promising ability of GFlowNets to generate high-quality and diverse candidates, they have achieved great success in challenging problems, including molecule discovery (Bengio et al., 2021; Li et al., 2022; Jain et al., 2023), biological sequence design (Jain et al., 2022), and causal modeling (Deleu et al., 2022; 2024a; Atanackovic et al., 2024).

Goal-directed learning has been well conceptualized and studied across various fields, particularly in reinforcement learning (Andrychowicz et al., 2017; Park et al., 2024; Niemueller et al., 2019; Addison, 2024), where it is known as goal-conditioned RL (GCRL) (Liu et al., 2022). GCRL trains a single model and learns general policies capable of reaching arbitrary target states, showcasing significant benefits and performance improvements. However, it remains largely unexplored in the context of GFlowNets with only a few prior studies. One such study proposes training goal-conditioned GFlowNets (GC-GFlowNets) (Pan et al., 2023a) that can reach any goal within the object space (Roy et al., 2023). It further demonstrates that GC-GFlowNets facilitate rapid adaptation to novel tasks with unseen rewards, eliminating the need to learn from scratch, unlike traditional GCRL.

---

[†]Correspondence to: Ling Pan (lingpan@ust.hk).

However, it is challenging to train GFlowNets conditioned on goals due to the sparse and binary nature of rewards, as the agent only receives positive rewards upon reaching the specified goals. GC-GFlowNets collect data by interacting with the environments in restricted steps, leading to a risk of getting trapped in constrained distributions (Yarats et al., 2021). Furthermore, previous approaches highly rely on the diversity and coverage of training trajectories, which poses a critical challenge when limited to offline data. This is particularly important in offline goal-conditioned learning, which enables training general goal-reaching policies from purely offline interaction trajectories without any further environment interaction (Ma et al., 2022a).

To tackle these challenges, we propose a novel approach called **R**etrospective **B**ackward **S**ynthesis (RBS), a simple yet effective method for efficiently training GC-GFlowNets that learns a unified forward policy capable of reaching any desired goals. Different from existing approaches inspired by GCRL literature (Pan et al., 2023a; Roy et al., 2023), RBS augments the forward trajectories collected by the forward policies with synthesized backward trajectories by looking backward, guided by the inherent backward policies. It is noteworthy that the synthesized backward trajectories represent successful experiences as they consistently reach the desired goals. The key insight of RBS lies in its data-driven approach, which enriches the training data with high *quality* and *diversity*. It not only transforms sequences of actions with failure rewards into successful experiences with meaningful rewards, thereby increasing the *quality* of training experiences, but also generates entirely novel trajectories to increase the *diversity* of data. Nevertheless, it is still challenging to scale it up to more complex and long-horizon problems, where GC-GFlowNets are prone to instabilities and mode collapse. We hypothesize this is due to ineffective reward gradient propagation caused by severe credit assignment issues, which results in poor utilization of valuable learning signals. To address these issues, we propose to intensify reward signals in the training objective to strengthen gradient backpropagation and regularize the backward policies to enhance the diversity of synthesized backward trajectories. We conduct a comprehensive evaluation of our method across various tasks from different benchmarks, where a binary bonus is awarded only if the agent reaches the desired goal. In comparison with a thorough set of baselines, our method largely improves sample efficiency during training and enhances the generalizability of GC-GFlowNets. Our contributions are three-fold:

- We present a novel method named Retrospective Backward Synthesis, which imagines a new trajectory from a desired goal, enhancing the quality and diversity of the training data.

- We introduce effective techniques, e.g., reward intensification and backward policy regularization, to stabilize and improve the training process.

- We showcase the effectiveness of our method through extensive experiments. A noteworthy result is that our method achieves about a 100% success rate in large-scale sequence generation tasks while all of the baselines completely fail. The results serve as a testament to its capability and underscore its potential for further GC-GFlowNets research.

## 2 PRELIMINARIES

### 2.1 GFLOWNETS

Let $\mathcal{X}$ denote the space of compositional objects and $R$ denote a reward function that assigns non-negative values to objects $x \in \mathcal{X}$. The non-negative reward function is denoted by $R(x)$. GFlowNets work by learning a sequential, constructive sampling policy $\pi$ that samples objects $x$ according to the distribution defined by the reward function ($\pi(x) \propto R(x)$). At each timestep, GFlowNets choose to add a building block $a \in \mathcal{A}$ (action space) to the partially constructed object $s \in \mathcal{S}$ (state space). This can be described by a directed acyclic graph (DAG) $\mathcal{G} = (\mathcal{S}, \mathcal{A})$, where $\mathcal{S}$ is a finite set of all possible states, and $\mathcal{A}$ is a subset of $\mathcal{S} \times \mathcal{S}$, representing directed edges. The generation of an object $x \in \mathcal{X}$ corresponds to a complete trajectory $\tau = (s_0 \to \cdots \to s_n) \in \mathcal{T}$ in the DAG starting from the initial state $s_0$ and terminating in a terminal state $s_n \in \mathcal{X}$. We define state flow $F(s)$ as a non-negative weight assigned to each state $s \in \mathcal{S}$. The forward policy $P_F(s'|s)$ is the forward transition probability over the children of each state, and the backward policy $P_B(s|s')$ is the backward transition probability over the parents of each state. The marginal likelihood of sampling $x \in \mathcal{X}$ can be derived as $P_F^\top(x) = \sum_{\tau=(s_0 \to \cdots \to x)} P_F(\tau)$. The primary objective of GFlowNets is to train a parameterized policy $P_F(\cdot|s, \theta)$ such that $P_F^\top(x) \propto R(x)$ (Bengio et al., 2021; 2023).

### 2.1.1 TRAINING CRITERIA OF GFLOWNETS

**Detailed Balance.** The detailed balance (DB) objective realizes the flow consistency constraint on the edge level, i.e., the forward flow for an edge $s \to s'$ matches the backward flow, as defined in Eq. (1). For terminal states $x$, it pushes $F(x)$ to match terminal rewards $R(x)$. DB learns to predict state flows $F_\theta(s)$, forward policy $P_F(\cdot|s; \theta)$ and backward policy $P_B(\cdot|s; \theta)$.

$$\forall s \to s' \in \mathcal{A}, \quad F_\theta(s)P_F(s'|s; \theta) = F_\theta(s')P_B(s|s'; \theta). \tag{1}$$

**(Sub-) Trajectory Balance.** Trajectory Balance (TB) extends DB from the edge level to the trajectory level based on a telescoping calculation of Eq. (1), which parameterized the normalizing constant $Z_\theta$, forward policy $P_F(\cdot|s; \theta)$ and backward policy $P_B(\cdot|s; \theta)$, whose learning objective is defined as $Z_\theta \prod_{t=1}^n P_F(s_t|s_{t-1}; \theta) = R(x) \prod_{t=1}^n P_B(s_{t-1}|s_t; \theta)$. Sub-trajectory Balance (SubTB) (Madan et al., 2023b) aims to mitigate the variance of TB and bias of DB, which considers the flow consistency criterion in the sub-trajectory level ($\tau_{i:j} = \{s_i \to \cdots \to s_j\}$), where $s_i$ and $s_j$ are not necessarily the initial and terminal state. The learning objective of SubTB for each sub-trajectory is defined as in Eq. (2). Its loss function is the squared difference between the left and right-hand sides of Eq. (2) (Madan et al., 2023a) in the log-scale, considering a weighted combination of all possible $O(n^2)$ sub-trajectories.

$$F_\theta(s_i) \prod_{t=i+1}^j P_F(s_t|s_{t-1}; \theta) = F_\theta(s_j) \prod_{t=i+1}^j P_B(s_{t-1}|s_t; \theta) \tag{2}$$

### 2.2 PROBLEM FORMULATION OF GC-GFLOWNETS

Inspired by the literature on goal-conditioned RL (Liu et al., 2022; Park et al., 2024; Veeriah et al., 2018), the idea of flow functions and policies in GFlowNets can be generalized to different goals $y$ in the goal space (Pan et al., 2023a), leading to the formulation of goal-conditioned GFowNets (GC-GFlowNets). GC-GFlowNets can be formulated as a goal-augmented DAG $\mathcal{G} = (\mathcal{S}, \mathcal{A}, \mathcal{Y}, \phi)$, where $\mathcal{Y}$ denotes the goal space describing the tasks, and $\phi : \mathcal{S} \to \mathcal{Y}$ is a tractable mapping function that maps the state to a specific goal. In this paper, we consider an identity function for $\phi$ following Pan et al. (2023a). In the goal-augmented DAG, the reward function $R(x, y) : \mathcal{S} \times \mathcal{Y} \to \mathbb{R}$ is also conditioned on goals, determining whether the goal object is reached:

$$R(x, y) = \begin{cases} 1, & \|\phi(x) - y\| \leq \epsilon \\ 0, & \text{otherwise} \end{cases}. \tag{3}$$

Therefore, the primary objective of GC-GFlowNets is to train a parameterized goal-conditioned forward policy $P_F(\cdot|s, y, \theta)$ such that $P_F^\top(x|y) \propto R(x, y)$, where $P_F^\top(x|y)$ is the marginal likelihood of sampling $x \in \mathcal{X}$ given $y$. Meanwhile, the flow function $F_\theta(s)$ and backward policy $P_B(s|s', \theta)$ can be extended to goal-conditioned flow and policy $F_\theta(s|y)$ and $P_B(s|s', y, \theta)$. Different from Eq. (1), the resulting learning objective for GC-GFlowNets for intermediate states is as follows:

$$\forall s \to s' \in \mathcal{A}, \quad F_\theta(s|y)P_F(s'|s, y, \theta) = F_\theta(s'|y)P_B(s|s', y, \theta). \tag{4}$$

In practice, GC-GFlowNets can be trained by minimizing the following loss function $\mathcal{L}$ in the log-scale (for numerical stability as discussed in Bengio et al. (2021)) as shown in Eq. (5), where $F_\theta(s'|y)$ is substituted with $R(s', y)$ if $s'$ is a terminal state.

$$\mathcal{L}_{\text{GC−GFN}} = \left( \log \frac{F_\theta(s|y)P_F(s'|s, y, \theta)}{F_\theta(s'|y)P_B(s|s', y, \theta)} \right)^2, \tag{5}$$

## 3 PROPOSED METHOD: RETROSPECTIVE BACKWARD SYNTHESIS

In this section, we first introduce a motivating example in §3.1 to demonstrate the insights and efficacy of our method intuitively. Subsequently, we introduce our novel approach and discuss the techniques we developed for improved efficiency in §3.2, which improves the training of GC-GFlowNets in a simple yet effective manner.

## 3.1 A MOTIVATING EXAMPLE

Training GC-GFlowNets according to Eq. (4) can be challenging due to the sparsity of reward signals – the agent receives a non-zero reward only when it reaches the desired goal state, while all other states yield zero reward. The high-dimensional space presents a further challenge, as the agent may spend a significant amount of time exploring unproductive, restricted regions of the state space without receiving any meaningful feedback. Reward relabeling (Andrychowicz et al., 2017; Fang et al., 2018; Pan et al., 2023a) aims to alleviate sparse reward issues in goal-conditioned tasks by relabeling the achieved states as goals, which have demonstrated success in the goal-conditioned RL literature, but still struggles to generalize to larger-scale scenarios with a large state space. This challenge is exacerbated in the context of GFlowNets, where agents can discover different trajectories leading to the same goal, and their number increases exponentially in dimensionality. Consequently, these reward labeling techniques may struggle to generalize across these different goal-trajectory relationships and are prone to get stuck in local optima, as they rely solely on observed data without expanding their data coverage.

We demonstrate this inefficiency problem in a goal-conditioned set generation task (Pan et al., 2023b), where the agent generates a set of size $|S|$ from $|U|$ distinct elements sequentially starting from an empty set. At each timestep, the agent chooses to add an element from $U$ to the current set $s$ (the GFlowNets state) without repeating elements. We randomly sample a target state of size $|S|$ from $U$ ($|U| = 30$) for each episode, and the GC-GFlowNets agent receives a negative reward of 0 as long as the final generated object is not the target state. Previous state-of-the-art method (Pan et al., 2023a) based on standard reward relabeling (HER (Andrychowicz et al., 2017)), struggles in this high-dimensional task for $|S| \geq 12$ with large state space.

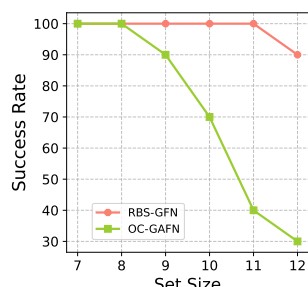

Figure 1: Succee rates with increasing set sizes in set generation.

These methods solely rely on experiences collected from interactions with environments, potentially trapping the agent in local optima and hindering the discovery of an effective goal-achieving strategy for problems with increasing scales.

## 3.2 PROPOSED METHOD

In this section, we propose a novel approach, Retrospective Backward Synthesis (RBS), which is a simple yet effective method to efficiently tackle these challenges mentioned above in GC-GFlowNets.

Consider a trajectory $\tau = \{s_0 \rightarrow \cdots \rightarrow s_i \rightarrow \cdots \rightarrow s_n\}$ collected by the forward policy $P_F$ of GC-GFlowNets that fails to reach the goal ($s_n \neq y$) and receives a zero reward. As illustrated in Fig. 2, RBS utilizes the potential of the backward policy $P_B$ to synthesize a backward trajectory $\tau' = \{y \rightarrow \cdots \rightarrow s_i' \rightarrow \cdots \rightarrow s_0\}$ from the commanded goal. When employing $\tau'$ for training, we reverse it to guarantee $\tau'$ starts from the initial state $s_0$, similar to $\tau$. Therefore, $\tau'$ provides a successful training experience as it achieves the goal state, thus enriching training data with positive feedback for GC-GFlowNets to mitigate the sparse reward problem. Unlike

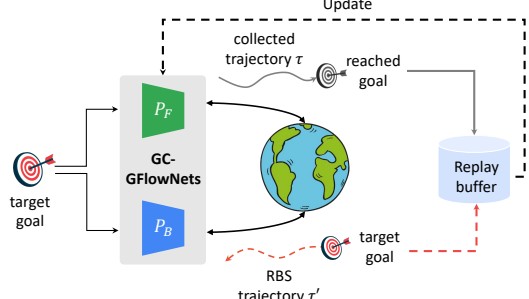

Figure 2: Overview of the Retrospective Backward Synthesis (RBS) approach.

previous reward relabeling techniques, such as HER (Andrychowicz et al., 2017), which simply replaces the original goal $y$ with the achieved state $s_n$, RBS provides more diverse and informative new training data in the trajectory level. RBS imagines a totally new trajectory $\tau' \neq \tau$, thus leading to significant data augmentation and more sample-efficient learning. In practice, we store both collected trajectories $\{\tau_i\}_{i=1}^m$ and experiences from RBS $\{\tau_i'\}_{i=1}^m$ in a replay buffer, and jointly replay the two types of trajectories to optimize GC-GFlowNets.

However, training GC-GFlowNets with RBS may still face the risks of learning instabilities and mode collapse when scaling to longer-horizon and more complex tasks, which may be caused by ineffective reward gradient backpropagation and inefficient experience replay. To address these challenges and

further enhance the diversity of generated backward trajectories, we introduce the techniques we have developed in the next paragraphs, where the overall algorithm is summarized in Alg. 1.

---

**Algorithm 1** Retrospective Backward Synthesis GFlowNets

---

1: **Initialize:** GC-GFlowNets $F_\theta(s|y)$, $P_F(s'|s, y, \theta)$ and $P_B(s|s', y, \theta)$ with parameters $\theta$, replay buffer $\mathcal{B}$, max priority $p_{max}$.
2: **for** $i = \{0, 1, \cdots, N - 1\}$ **do**
3:     Sample a goal $y \sim \mathcal{Y}$ randomly
4:     Collect a forward trajectory $\tau = \{s_0 \rightarrow s_1 \rightarrow \cdots \rightarrow s_n\}$ with $P_F$, and obtain reward $R(s_n, y) \leftarrow \mathbb{I}\{(s_n = y\}$
5:     Store $\mathcal{T} = (\tau, y, R)$ with priority $p_{\max} > 0$ in $\mathcal{B}$
6:     Collect a backward trajectory $\tau' = \{y \rightarrow \cdots \rightarrow s'_1 \rightarrow s_0\}$ with retrospective backward synthesis using $P_B$, and obtain reward $R(y, y) \leftarrow \mathbb{I}\{(y = y\} \equiv 1$
7:     Store $\mathcal{T}' = (\tau', y, R)$ with priority $p_{\max} > 0$ in $\mathcal{B}$
8:     Sample a batch $\{\mathcal{T}_i\}_{i=1}^m$, $\{\mathcal{T}'_i\}_{i=1}^m$ proportionally to their priorities from $\mathcal{B}$.
9:     Update GC-GFlowNet towards minimizing Eq. (6)
10:    Update priorities of $\mathcal{T}$ and $\mathcal{T}'$, and the weighting coefficient $\gamma$
11: **end for**

---

**Age-Based Sampling.** Trajectories collected by GC-GFlowNets and the retrospective backward synthesized experiences are stored in a reply buffer for better data utilization. However, uniform sampling of these goal-reaching trajectories often fails to expose the agent to sufficiently diverse experiences that match its evolving learning ability (Fang et al., 2019), which highlights the importance of sampling strategy (Kloek & van Dijk, 1976; Schaul et al., 2016). To guarantee that all experiences are fully considered during training, we introduce an age-based sampling technique. Specifically, age-based sampling assigns the highest priority $p_{\max} > 0$ to newly added experiences and updates their priority to zero after being learned. Consequently, newly added experiences can be replayed first, while learned experiences are randomly sampled. This prioritization scheme ensures that experiences are leveraged more thoroughly, which balances the exploration of fresh experiences and the exploitation of acquired knowledge.

**Backward Policy Regularization.** The backward policy can be chosen freely as studied in (Bengio et al., 2023). In the extreme case where $P_B$ is set to be a uniform policy, the optimization of GC-GFlowNets becomes challenging as it is not learnable and the data is excessively diverse. On the other hand, specifying $P_B$ as a deterministic policy can limit data diversity. We therefore learn the backward policy $P_B(\cdot|s', \theta)$ within GC-GFlowNets based on the flow consistency criterion in §2.2, instead of specifying it to be a fixed policy, which offers smooth sampling aligned with the current model's capacity. Yet, when $P_B$ degenerates into a deterministic policy, it fails to provide diverse backward trajectories, which can limit the potential of GC-GFlowNets to generalize well. To strike a balance between these two extremes and further enhance the diversity of the imagined trajectories, we introduce a backward policy regularization for $P_B$. This regularization term penalizes the Kullback-Leibler (KL) divergence between $P_B(\cdot|s', \theta)$ and a uniform distribution $\mathcal{U}$, thus encouraging $P_B$ to resemble a uniform distribution, while allowing for learning and adaptation. Our training objective can be written as in Eq. (6), where $\gamma$ is the regularization coefficient.

$$\mathcal{L}_{\text{RBS-GFN}} = \mathcal{L}_{\text{GC-GFN}} + \gamma \times D_{\text{KL}}(P_B(\cdot|s', y, \theta)\|\mathcal{U}). \tag{6}$$

To avoid interference with the original training objective $\mathcal{L}_{\text{GC-GFN}}$, we employ a linearly decaying hyperparameter $\beta$ to regulate the coefficient $\gamma$ (i.e., $\gamma \leftarrow \beta \times \gamma$). Consequently, the KL penalty gradually diminishes towards zero over the course of training.

**Intensified Reward Feedback.** For long-horizon and high-dimensional tasks, a critical factor that affects learning effectiveness is the efficient propagation of the reward signal, which may require a number of steps and affect the learning of intermediate steps. The recent OC-GAFN (Pan et al., 2023a) approach considers the terminal reward at each step (due to the binary nature of rewards), but may introduce stochasticity of the learning signal particularly in the case of the more challenging graph-structured DAG. We propose an efficient technique that intensifies the learning signal, defined as $\mathcal{L}_{\text{GC-GFN}} = (\log [F_\theta(s|y)P_F(s'|s, y, \theta)] - \log [CR(x, y)P_B(s|s', y, \theta)])^2$ for terminal states $s'$, where $C$ is a intensification coefficient to scale the effect of $R(x, y)$. The mechanism behind this technique is that a larger value of $C$ indeed amplifies the gradient of $P_B$ by $\log(CR(x, y))$ for terminal states (the detailed derivation can be found in Appendix A). By the intensified reward feedback with

a large value of $C$, we effectively strengthen the learning signal propagated backward through the trajectory, enabling more efficient learning and faster convergence in complex environments.

While our proposed method can effectively improve the training of GC-GFlowNets, it may still face challenges when dealing with extremely large-scale state spaces, e.g., antimicrobial peptides generation (Malkin et al., 2022) with $20^{50}$ possible states. To further improve its scalability, we propose a hierarchical approach for RBS that decomposes the task into low-level sub-tasks that are easier to complete. Detailed descriptions for how we realize hierarchical decomposition for RBS can be found in Appendix B due to space limitation.

**Empirical Validation** Fig. 1 compares RBS and OC-GAFN (with HER (Andrychowicz et al., 2017)) in the set generation task with increasing set sizes. The result demonstrates that our RBS approach can scale up to problems with large set sizes, while the previous SOTA method OC-GAFN fails to maintain good performance as the problem complexity increases.

## 4 RELATED WORK

**Generative Flow Networks (GFlowNets).** Recently, there have been a number of efforts applying GFlowNets to different important cases, e.g., biological sequence design (Jain et al., 2022), molecule generation (Bengio et al., 2021), combinatorial optimization (Zhang et al., 2023a; 2024), Bayesian structure learning (Deleu et al., 2022). There have also been many works investigating how to improve the training of GFlowNets, enabling them to achieve more efficient credit assignment (Pan et al., 2023b), better exploration (Pan et al., 2023c; Lau et al., 2024), or more effective learning objectives (Bengio et al., 2023; Madan et al., 2023a) that can better handle computational complexity (Bengio et al., 2021), and generalize to stochastic environments (Pan et al., 2023d; Zhang et al., 2023b). GC-GFlowNets learn flows and policies conditioning on outcomes (goals) for reaching any targeted outcomes (Pan et al., 2023a). However, little attention has been given to this topic, leaving this promising direction largely unexplored. Meanwhile, it is challenging to train goal-conditioned policies due to sparse rewards. Our work not only provides a formal definition of GC-GFlowNets but also proposes a novel method called retrospective backward synthesis to significantly improve their training efficiency and success rates.

**Goal-Conditioned Reinforcement Learning.** Our formulation of goal-conditioned GFlowNets is heavily inspired by the works of goal-conditioned RL. Standard Reinforcement Learning (RL) only requires the agent to finish one specific task defined by the reward function (Schaul et al., 2015), while goal-conditioned RL trains an agent to achieve arbitrary goals as the task specifies (Andrychowicz et al., 2017). Goal-Conditioned RL augments the observation with an additional goal that the agent is required to achieve (Liu et al., 2022). The reward function is usually defined as a binary bonus of reaching the goal. To overcome the challenge of the sparsity of reward function, prior work in goal-conditioned RL has introduced algorithms based on a variety of techniques, such as hindsight relabeling (Andrychowicz et al., 2017; Fang et al., 2018; Yang et al., 2022; Fang et al., 2019; Ding et al., 2019), contrastive learning (Eysenbach et al., 2020; 2022), state-occupancy matching (Durugkar et al., 2021; Ma et al., 2022b) and hierarchical sub-goal planning (Chane-Sane et al., 2021; Kim et al., 2021; Nasiriany et al., 2019). Our work is closely related to hindsight relabeling, denoted as HER (Andrychowicz et al., 2017), which relabels any experience with some commanded goal to the goal that was actually achieved in order to learn from failures. HER can generate non-negative rewards to alleviate the negative sparse reward problem, even if the agent did not complete the task. However, the agent using HER still suffers from low sample efficiency on large-scale problems due to its limitation in operating only on the observed trajectories, while our method can imagine new trajectories with positive rewards for policy training.

## 5 EXPERIMENTS

In this section, we conduct extensive experiments to investigate our Retrospective Backward Synthesis (RBS) method to answer the following key questions: i) How does RBS-GFN compare against previous baselines in terms of sample efficiency and success rates? ii) Can RBS-GFN scale to complex and high-dimensional environments? iii) Can RBS-GFN effectively generalize to unseen goals and unseen environments? iv) Is RBS-GFN general and can be built upon different GFlowNets training objectives? v) What are the effects of important components in RBS-GFN?

## 5.1 GRIDWORLD

We first conduct a series of experiments based on the GridWorld environment (Bengio et al., 2021), in which the model learns to achieve any given goals starting in a $H \times H$ grid. Specifically, we investigate mazes with increasing horizons $H$ (32, 64, and 128), respectively, resulting in different levels of difficulty categorized as *small*, *medium*, and *large*.

We compare our proposed RBS-GFN approach with the following state-of-the-art baselines. (i) GFN w/ HER (Andrychowicz et al., 2017) is a GC-GFlowNets that relabels the negative reward in a failed trajectory with a positive reward. (ii) OC-GAFN (Pan et al., 2023a) is a recent method that utilizes contrastive learning to complement successful experiences, and employs a trained Generative Augmented Flow Network (GAFN; Pan et al. (2023c)) as an exploratory component to generate diverse outcomes $y$, which are subsequently provided to sample goal-conditioned trajectories. (iii) DQN w/ HER leverages both deep Q-learning algorithm (Mnih et al., 2013; Jang et al., 2019) and HER technique (Andrychowicz et al., 2017) to learn a near-optimal policy. This baseline is used to ablate the effects of GFlowNet-based training compared with RL-style methods. To ensure fairness, each baseline has the same model architecture and training steps as RBS-GFN, and we follow the experimental setup for hyperparameters as in (Pan et al., 2023a). We run each algorithm with three different seeds and report their performance in mean and standard deviation. A more detailed description of the experimental setup can be found in Appendix C.

### 5.1.1 PERFORMANCE COMPARISON

The success rates for different methods for increasing sizes of the GridWorld environment (including small, medium, and large) are summarized in Fig. 3. We obtain the following observations based on the results. (i) GFN-based goal-conditioned approaches consistently outperform RL-based goal-conditioned methods (DQN w/ HER), as the latter can easily get trapped in local optima due to its greedy policy. The results validate the promise of training goal-conditioned policies using GFlowNets and pave the way for further advancements in goal-conditioned learning with GFlowNets. (ii) Moreover, our proposed RBS-GFN method significantly outperforms GFN w/ HER and is stronger than the OC-GAFN method in terms of sample efficiency and outcome-reaching ability, particularly in larger spaces. In contrast, the performance of GFN w/ HER deteriorates as the complexity of the environment increases, which highlights its limitations in handling large state spaces. (iii) The inferior performance of our method without RBS highlights the significance of our proposed approach. We remark that the superior performance of RBS is attributed to its enhancement of training data with higher quality and diversity.

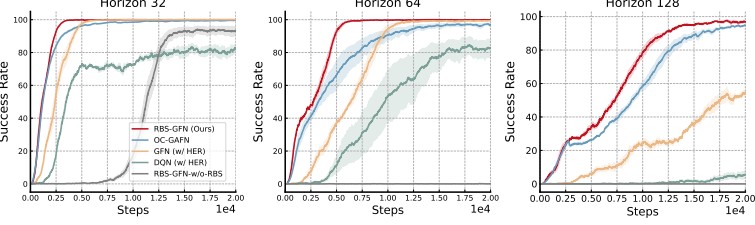
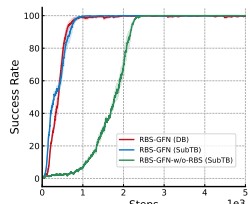

Figure 3: Performance comparison in GridWorld. *Left*: Small. *Middle*: Medium. *Right*: Large.

Figure 4: Results of RBS-GFN (SubTB).

### 5.1.2 SPECIAL CASE: OFFLINE GC-GFLOWNETS

Given the potential of learning general goal-reaching policies solely from given offline datasets Ma et al. (2022a), we investigate the offline goal-conditioned scenario where GC-GFlowNets learn from fixed offline data without interacting with the environments. As a result, all the baselines are restricted in limited training datasets, while RBS-GFN can synthesize a number of new trajectories to enhance the learning process. We compare our method with the two strongest baselines, OC-GAFN and vanilla goal-conditioned GFN, and evaluate them on three different sizes of GridWorld. As demonstrated in Fig. 5, our method achieves nearly 100% success rates across all scenarios, while the performance of the baselines declines significantly as the problem size increases. The experimental details and learning curves can be found in Appendix C.3.

### 5.1.3 GENERALIZATION

We now evaluate the generalization ability of our RBS-GFN method to unseen goals and environments, which is important for real-world applications where the agent can encounter novel situations. To evaluate its ability to generalize to unseen goals, we mask $n$ goals from various locations in the map and test the success rates of reaching these unseen goals after the training process, as illustrated in Fig. 6(a) ($n = 20$). As shown in Fig. 6(b), RBS-GFN obtains an almost 100% success rate, demonstrating its capacity to effectively determine the required actions to reach novel goals. Moreover, it outperforms our strongest baseline OC-GAFN by an approximately 15% success rate. We further investigate its generalization capability to unseen environments. We introduce

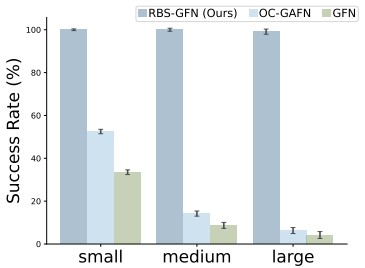

Figure 5: Success rates on Grid-World tasks with different sizes.

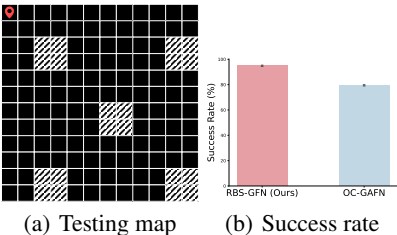

(a) Testing map    (b) Success rate

Figure 6: (a) Visualization of GridWorld. 📍 is the start point, and ▨ is the unseen goal. (b) The average success rate of reaching these unseen goals for 100 trials per goal.

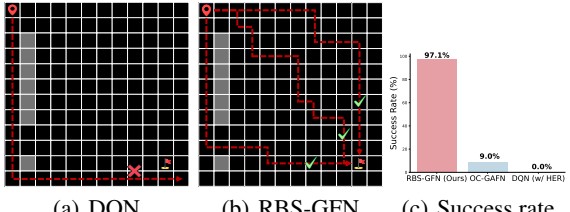

(a) DQN    (b) RBS-GFN    (c) Success rate

Figure 7: (a) DQN fails to generalize to unseen maps with obstacles. (b) RBS-GFN can find diverse trajectories. (c) The average success rate over 200 trials of reaching the goal 🚩

unseen obstacles during the testing phase following (Kumar et al., 2020), which creates novel environments that the agent has not encountered during training. As shown in Fig. 7(c), RBS-GFN maintains a success rate of almost 100%, while OC-GFN obtains a success rate of 9% and the RL-based method DQN completely fails. More unseen maps and corresponding results are provided in Appendix D. The superior performance of RBS-GFN in unseen environments can be attributed to its ability to efficiently discover diverse paths to reach the goal as shown in Fig. 7(b). Although OC-GAFN also has the potential to discover diver paths, its performance is limited by the available training budget (i.e., fewer pre-training iterations). On the other hand, DQN is limited to discovering a single trajectory to reach the goal as shown in Fig. 7(a), making it highly susceptible to failure when the learned path is blocked by unseen obstacles.

### 5.1.4 VERSATILITY

In this section, we demonstrate the generality of our approach by integrating it with another recent GFlowNets method based on SubTB (Madan et al., 2023b), whose learning objective is based on $\mathcal{L}_{\text{SubTB}}$ as introduced in Eq. (2). We evaluate the goal-reaching performance of RBS-GFN (SubTB) in terms of the success rate in the GridWord task (with $H = 10$).

As shown in Fig 4, RBS-GFN can also be successfully built upon SubTB with a success rate of 100%, and achieves consistent performance gains.

### 5.1.5 ABLATION STUDY

In this section, we conduct an in-depth analysis of the key components of RBS-GFN to better understand their effect with a focus on two critical techniques, including backward policy regularization and age-based sampling, while we defer the discussion of intensified reward feedback, which is essential for scaling up to high-dimensional problems in Appendix A.1.

The backward sampling policy $P_B$ plays an important role in synthesizing helpful trajectories for training GC-GFlowNets. We evaluate the effect of different choices of $P_B$, including the regularized $P_B$ (based on Eq. (6)), a learned backward policy without constraints, and a fixed and uniform one. We compute the entropy of the forward policy $P_F$ to measure the ability to generate diverse trajectories of GC-GFlowNets. We further visualize the trajectory distribution in the replay buffer for different choices of $P_B$ with t-SNE (Van der Maaten & Hinton, 2008) in Fig. 8(b).

As shown in Fig 8(a), the proposed regularized $P_B$ converges faster than other variants in terms of success rate while maintaining a satisfactory level of entropy. Furthermore, Fig 8(b) illustrates that the synthesized trajectory distribution of regularized $P_B$ and uniform $P_B$ cover a wide range, which effectively compensates for the limited coverage of the original data distribution, while learned $P_B$ struggles to synthesis trajectories that significantly differs from the original data distribution.

Fig. 10 demonstrates the effect of our proposed age-based sampling technique (with horizon $H = 128$), which highlights its importance in improving learning efficiency and stability, as the model struggles to efficiently achieve a high success rate without age-based sampling (which fails to fully utilize and learn from newly-generated samples).

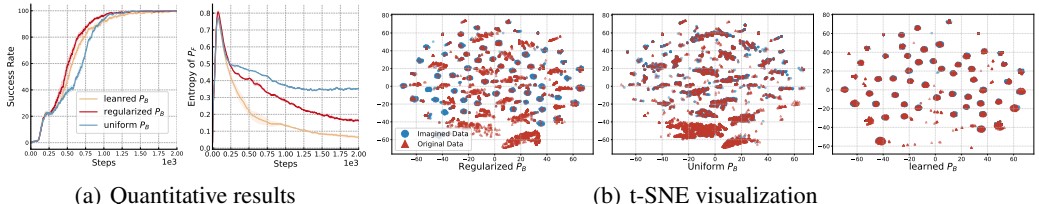

(a) Quantitative results  (b) t-SNE visualization

Figure 8: Comparison results of using different $P_B$ to synthesize experiences.

## 5.2 BIT SEQUENCE GENERATION

In this section, we investigate the performance of RBS-GFN in the bit sequence generation task (Malkin et al., 2022). Unlike previous approaches that generate these sequences in a left-to-right manner (Malkin et al., 2022; Madan et al., 2023a), we adopt a non-autoregressive prepend/append Markov decision process following Shen et al. (2023), which is a more challenging task compared to the one studied in Pan et al. (2023a). The action space includes pretending or appending a $k$-bit word from the vocabulary $V$ to the current state, which increases the difficulty of the task (as the underlying structure of the problem is a directed acyclic graph rather than a simple tree (Malkin et al., 2022)). We consider bit sequence generation with small, medium, and large sizes with increasing lengths and vocabulary sizes following Pan et al. (2023a).

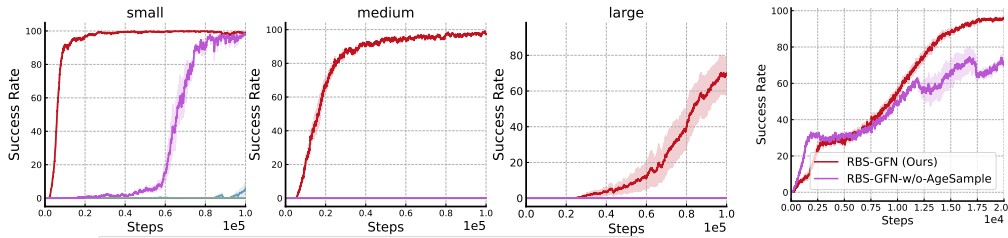

Figure 9: Performance comparison in bit sequence generation.

Figure 10: Performance compared with RBS-GFN-w/o-AgeSample.

As shown in Fig. 9, even the strongest OC-GAFN method struggles to learn efficiently given a limited training budget, while all other baselines completely fail. In contrast, RBS-GFN achieves high success rates of approximately 100% with fast convergence across different scales of the tasks. It is worth noting that RBS-GFN without either age-based sampling (denoted as RBS-GFN-w/o-AgeSample) or intensified reward feedback (see detailed discussions in Appendix A.1) both fail to generalize to more complex tasks, including *medium* and *large*, demonstrating the importance of our proposed techniques in enabling RBS-GFN to efficiently learn across various levels of complexity.

## 5.3 TF BIND GENERATION

In this section, we study a more practical task of generating DNA sequences with high binding activity with targeted transcription factors (Jain et al., 2022). Similar to the bit sequence generation task, the agent prepends or appends a symbol from the vocabulary to the current state at each step. As shown in Fig. 11(a), RBS-GFN archives a success rate of $100\%$ and learns much more efficiently thanks to its retrospective backward synthesis mechanism, and outperforms other baselines, which illustrates its effectiveness for DNA sequence generation. We provide additional experimental analysis about this task in Appendix D.

## 5.4 AMP GENERATION

In this section, we study the antimicrobial peptides (AMP) (Jain et al., 2022) generation task for investigating the scalability of our proposed method. The task involves generating a sequence with a length of 50 from a vocabulary with a size of 20. We follow the same experimental setup as in §5.3, considering an action space with prepend and append operations following Shen et al. (2023). The state space contains $20^{50}$ possible AMP sequences, which poses a significant challenge for efficient exploration and optimization. Moreover, the task is extremely difficult due to the vast sequence space and complex structure-function relationships compared to the case studied in Pan et al. (2023a).

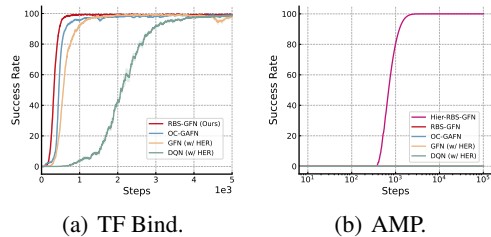

(a) TF Bind.          (b) AMP.

Figure 11: Succee rates on the TF Bind and AMP sequence generation tasks.

To tackle this large-scale problem, we employ the goal decomposition method proposed in §3.2 (see details in Appendix B. By breaking down the target goal into simpler sub-goals, i.e., generating a shorter sub-sequence, we can effectively reduce the complexity of the search space, enabling more efficient learning. We refer to this approach as Hier-RBS-GFN. As demonstrated in Fig. 11(b), Hier-RBS-GFN significantly improves the learning efficiency and outperforms all baseline methods, which shows the scalability of our approach for tackling complex tasks with vast search spaces.

## 5.5 APPLICATION: DOWNSTREAM FINETUNING

A notable advantage of GC-GFlowNets is that the pre-trained policy can be leveraged to handle downstream tasks with unseen rewards, unlike the typical fine-tuning process of reinforcement learning as they generally learn reward-maximization policies that may discard valuable information (Pan et al., 2023a). In this section, we study the application of GC-GFlowNets and validate its effectiveness for adapting to downstream bit sequence generation tasks with unseen rewards in different scales. We generally follow the experimental design in Pan et al. (2023a), while we train GC-GFlowNets with fewer pre-training budgets, which better illustrates the efficiency of our method in the pre-training stage. From the results shown in Fig. 12, we find that RBS-GFN outperforms OC-GAFN by a large margin, as a more efficient and effective pre-trained goal-reaching strategy can significantly contribute to the fine-tuning process, while OC-GAFN struggles to efficiently discover modes given limited pre-training budgets in this challenging goal-conditioned training stage.

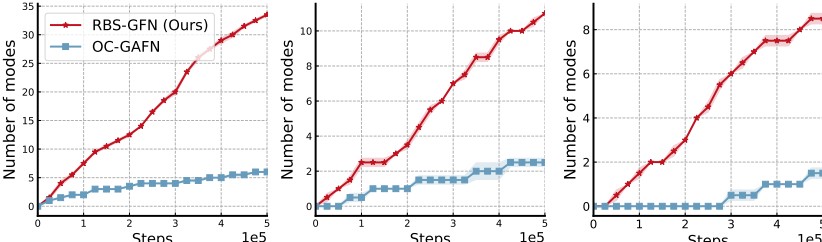

Figure 12: Fine-tuning GC-GFlowNets on the downstream sequence generation task with different scales. We report the number of modes discovered during training. The number of modes is calculated using a sphere-exclusion procedure. A candidate is added to the list of modes if it is above a certain reward threshold and is further away than some distance threshold from all other modes.

## 6 CONCLUSION

In this paper, we address the critical challenges of realizing goal-directed behavior and learning in GFlowNets. To overcome the training challenge due to extremely sparse rewards, we propose a novel method called Retrospective Backward Synthesis, which significantly improves the training of goal-conditioned GFlowNets by synthesizing backward trajectories. Our extensive experiments demonstrate state-of-the-art performance in terms of both success rate and generalization ability, which outperforms strong baselines. For future work, it is promising to further improve our method, e.g., sampling method considering alternative priorities (Sujit et al., 2023). More discussions about our work are provided in Appendix E.

## ACKNOWLEDGMENTS

This work of Haoran He and Ling Pan is supported by National Natural Science Foundation of China 62406266. Huazhe Xu is supported by Tsinghua dushi program. We also thank the anonymous reviewers for their valuable suggestions.

## ETHICS STATEMENT

This paper presents work whose goal is to advance the field of Machine Learning. Specifically, we propose a novel method called RBS to enhance the learning of GC-GFlowNets. Since this method is easy to reproduce (as we will release our code soon) and exhibits the SOTA performance, it encourages future research to further advance this field. There are many potential societal consequences of our work, none of which we feel must be specifically highlighted here.

## REPRODUCIBILITY STATEMENT

All details of our experiments can be found in Appendix C, which includes descriptions of the tasks, experimental setup, network architecture, and hyperparameters. The proof of intensified reward feedback is referred to in Appendix A. The code will be open-sourced upon publication of this work.

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

# A    INTENSIFIED REWARD FEEDBACK

We re-write the loss function of GC-GFlowNets in the case if $s'$ is terminal as follows:

$$\mathcal{L}_{\mathrm{GC-GFN}} = \left(\log F_\theta(s|y) P_F(s'|s, y, \theta) - \log \left[CR(x, y)P_B(s|s', y, \theta)\right]\right)^2, \tag{7}$$

which can be degenerated to Eq. (5) when we set $C = 1$. In practice, we set $C$ to a large value to facilitate effective reward propagation. Below, we demonstrate that a large $C$ scales the gradient with respect to $P_B$ without affecting $P_F$ or $F$. We show that

$$\frac{\partial \mathcal{L}_{\mathrm{GC-GFN}}}{\partial \theta} = 2 \times Z \frac{\partial \log Z}{\partial \theta}, \tag{8}$$

where

$$Z = \log F_\theta(s|y) + \log P_F(s'|s, y, \theta) - \log \left[CR(x, y)P_B(s|s', y, \theta)\right], \tag{9}$$

$$\frac{\partial \log Z}{\partial \theta} = \frac{1}{F_\theta(s|y)} \frac{\partial F_\theta(s|y)}{\partial \theta} + \frac{1}{P_F(s'|s, y, \theta)} \frac{\partial P_F(s'|s, y, \theta)}{\partial \theta} - \frac{1}{CR(x, y)P_B(s|s', y, \theta)} \frac{\partial \left(CR(x, y)P_B(s|s', y, \theta)\right)}{\partial \theta}. \tag{10}$$

Since the scaling coefficient $C$ does not depend on the model parameters $\theta$, the derivative w.r.t. $\theta$ simplifies as follows:

$$\frac{\partial \left(CR(x, y)P_B(s|s', y, \theta)\right)}{\partial \theta} = CR(x, y) \frac{\partial P_B(s|s', y, \theta)}{\partial \theta}. \tag{11}$$

Substituting this into Eq. 10, we obtain:

$$\frac{\partial \log Z}{\partial \theta} = \frac{1}{F_\theta(s|y)} \frac{\partial F_\theta(s|y)}{\partial \theta} + \frac{1}{P_F(s'|s, y, \theta)} \frac{\partial P_F(s'|s, y, \theta)}{\partial \theta} - \frac{1}{P_B(s|s', y, \theta)} \frac{\partial P_B(s|s', y, \theta)}{\partial \theta}, \tag{12}$$

where $C$ is eliminated. Consequently, $C$ only appears in the last term of $Z$. Therefore it only affects $P_B$, leaving gradients w.r.t. $P_F$ and $F_\theta$ unchanged.

## A.1    EMPIRICAL VALIDATION

To validate the effects of our proposed technique, i.e., intensified reward feedback, we conduct the ablation study in the bit sequence generation tasks, which are more complex and high-dimensional than the GridWorld tasks. In practice, we set $C = 1e^7$ for *small* task, $C = 1e^{25}$ for *medium* task, and $C = 1e^{40}$ for *large* task. From the results shown in Fig 13, we observe that RBS-GFN completely fails in the task without intensified reward feedback, obtaining only a $0\%$ success rate.

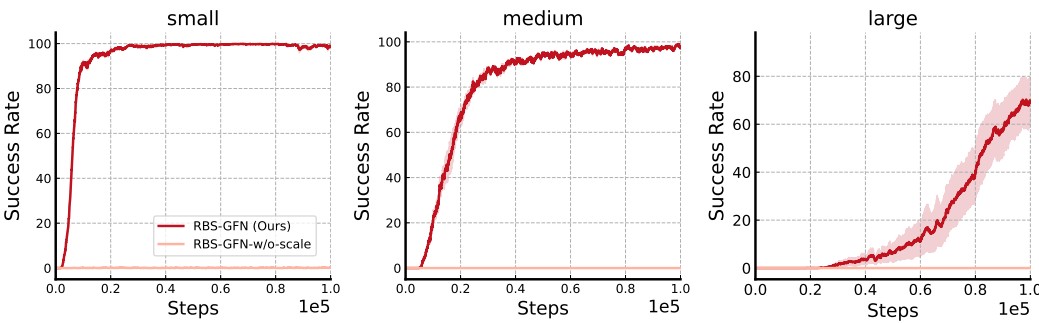

Figure 13: Performance comparison with RBS-GFN without using intensified reward feedback.

## B    HIERARCHICAL GOAL DECOMPOSITION

It is still challenging to tackle problems with extremely large-scale state spaces, and even our method can fail in these scenarios. To address this problem, we propose a hierarchical method to decompose the task into several low-level tasks that are easier to complete. Leveraging the consistent sequential structure in compositional GFlowNets tasks, we can set sub-goals manually, eliminating the need to learn a sub-goal generation policy additionally (Chane-Sane et al., 2021; Kim et al., 2021; Nasiriany et al., 2019). After training sub-level policies, all the generated sub-goals can be combined together to obtain the final goal.

Considering a sequence generation problem (Jain et al., 2022) as an example, wherein an agent is tasked with generating a sequence of length $l$ from a vocabulary of size $|\mathcal{V}|$, we can decompose this task into $k$ sub-level tasks. Consequently, we can train $k$ models, each capable of generating a sequence of length $l/k$. Subsequently, the $k$ generated sub-sequences can be concatenated to form a sequence of length $l$.

## C    EXPERIMENTAL SETUP

We build our implementation for all baselines and environments upon publicly available open-source repositories.[1] The code will be open-sourced upon publication of the work.

**GridWorld.**    The GridWorld (Bengio et al., 2021) is conceptualized as a 2-dimensional hypercube with side length $H : \{(s^1, s^2)|s^i \in \{0, 1, \cdots, H - 1\}$, where the model learns to achieve any given goals (outcomes) starting from a fixed initial state $(0, 0)$. We examine grids with $H$ set to 32, 64, and 128, respectively, resulting in different levels of difficulty categorized as *small*, *medium*, and *large*. The agent receives a positive reward of 1 only if it reaches the desired goal state. We use the Adam (Kingma & Ba, 2014) optimizer with a learning rate of $1e^{-3}$ for $2e^4$ training steps.

**Bit Sequence Generation.**    This task requires the model to generate sequences of length $n$ by pretending or appending a $k$-bit word to the current state. We consider $k = 2, n = 40$ for *small* task, $k = 3, n = 60$ for *medium* task, and $k = 5, n = 100$ for *large* one. We use the Adam (Kingma & Ba, 2014) optimizer with a learning rate of $5e^{-4}$ for $1e^5$ training steps.

**TF Bind Generation.**    Similar to the bit sequence generation task, the agent prepends or appends a symbol from the vocabulary with a size of 4 to the current state at each step to generate a sequence of length 8. We use the Adam (Kingma & Ba, 2014) optimizer with a learning rate of $5e^{-4}$ for $5e^3$ training steps.

**AMP Generation.**    This biological task requires the agent to generate antimicrobial peptides (AMP) with lengths of 50 (Jain et al., 2022) from a vocabulary with size of 20. For both RBS-GFN, Hier-RBS-GFN and all the baselines, we use the Adam (Kingma & Ba, 2014) optimizer with a learning rate of $5e^{-4}$ for $1e^5$ training steps.

### C.1    IMPLEMENTATION DETAILS

We describe the implementation details of our method as follows:

- We use an MLP network that consists of 2 hidden layers with 2048 hidden units and ReLU activation (Xu et al., 2015).
- The trajectories are sampled from a parallel of 16 rollouts in the environment at each training step.
- We set the replay buffer size as $1e6$ and use a batch size of 128 for sampling data and computing loss function.
- We combine the current state and goal state together as the input of our model. The input is transformed as one-hot embedding followed by our MLP model.
- We run all the experiments in this paper on an RTX 3090 machine.

---

[1] `https://github.com/GFNOrg/gflownet`

### C.2 BASELINES

We describe the implementation details of the baselines we use throughout this paper as follows:

- The only difference between **GFN w/ HER** and our method is that GFN w/ HER leverages HER (Andrychowicz et al., 2017) technique to enhance training experiences, while we utilize our proposed retrospective backward synthesis to augment the data with new reverse trajectories.

- For **OC-GAFN**, we follow the same experimental setup described in (Pan et al., 2023a). This method not only leverages goal relabeling (Andrychowicz et al., 2017) but also uses GAFN (Pan et al., 2023c) to generate diverse outcomes $y$, which are subsequently provided to sample outcome-conditioned trajectories. OC-GAFN requires training an additional GAFN model, which would be computationally expensive. At each training step, OC-GAFN takes 2 times gradient update, where one is for the negative samples and the other is for the relabeled samples. OC-GAFN does not maintain a replay buffer and uses newly sampled data to train its model.

- Following the implementation in (Andrychowicz et al., 2017), **DQN w/ HER** leverages both deep Q-learning algorithm (Mnih et al., 2013; Jang et al., 2019) and HER technique (Andrychowicz et al., 2017) to learn a near-optimal policy.

- **SAC w/ HER** leverages both SAC algorithm (Haarnoja et al., 2018) and HER technique (Andrychowicz et al., 2017) to learn a near-optimal entropy-regularized policy. We follow the hyperparameters used in (Huang et al., 2022).

### C.3 DETAILS OF OFFLINE EXPERIMENTS

For offline dataset collection, we store the samples recorded in the replay buffer of vanilla GFN during training until convergence. This dataset includes 8,016 trajectories with varying levels of performance. Given that the training of GFlowNets is off-policy, we reuse the learning objective in Eq. 6 without modification. The implementation details of our offline algorithm (RBS-GFN) and the baselines are illustrated below:

- (offline-) **RBS-GFN** (ours): Similar to the online version of RBS-GFN, the dataset is augmented by the reverse trajectories collected by $P_B$ at each training step.

- (offline-) **OC-GAFN**: Unlike RBS-GFN, OC-GAFN fails to generate new trajectories. Instead, it only augments the dataset by relabeling the outcomes of failed trajectories with their actual terminal states.

- (offline-) **GFN**: This baseline utilizes the original dataset without any additional data generation or augmentation processes.

To ensure a fair comparison, all other settings, including the network architecture and hyperparameters, are kept the same as those used in the online implementations.

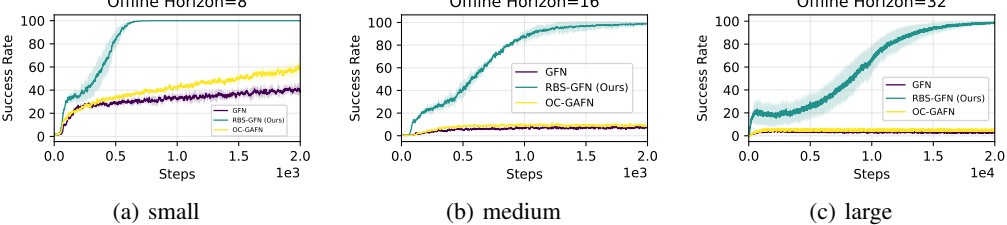

Figure 14: Learning curves of offline experiments on three different scales of GridWorld. Success rates are averaged over three random seeds.

## D ADDITIONAL EXPERIMENTAL RESULTS

We provide more experimental results that demonstrate the superior ability of our method.

### D.1 COMPUTATION OVERHEAD

RBS-GFN is efficient since we only need to synthesize a single $\tau'$ using backward policy $P_B$ from the goal, and rollout a minibatch of data at each training step. It is also worth noting that the strongest

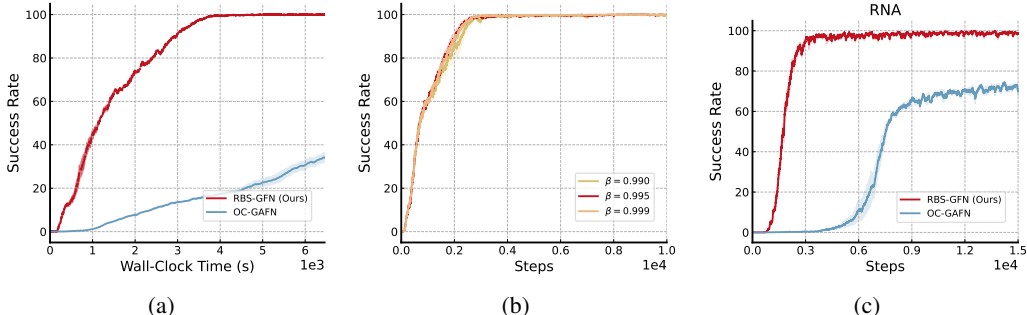

Figure 15: Additional experimental results. (a) Success rates on GridWorld with a horizon of 32. The x-axis corresponds to the wall clock time. (b) Ablation study with different values of the decay hyperparameter $\beta$. RBS-GFN is robust to different $\beta$. (c) Success rates on RNA generation (Pan et al., 2023a). RBS-GFN consistently outperforms the strongest baseline, OC-GAFN, on the task.

baseline OC-GAFN requires training two GFN models (i.e., an unconditioned GAFN model and a goal-conditioned GFN model), while RBS-GFN only needs to train a single goal-conditioned GFN agent, which largely reduces training compute requirements. We quantify the wall-clock time and corresponding achieved success rates on the GridWorld task with a horizon of 32. From the results shown in Fig. 15(a), we find that RBS-GFN achieves a significantly higher success rate in less time compared to the previous strongest method, OC-GAFN.

### D.2 RESULTS ON THE RNA GENERATION TASK

We further evaluate the performance of RBS-GFN on the RNA generation task (Lorenz et al., 2011), which involves constructing RNA sequences following Pan et al. (2023a). We compare our method with the strongest baseline, OC-GAFN. From the results shown in Fig. 15(c), we observe that RBS-GFN consistently achieves the best performance.

### D.3 ROBUSTNESS TO HYPERPARAMETERS

Regarding the reward intensification technique, the scaling coefficient $C$ is the only task-dependent hyperparameter that requires tuning, as it depends on the nature of specific tasks and also accommodates different horizons. However, this can be easily done through standard techniques like grid search (similar to tuning conventional hyperparameters such as the learning rate (Malkin et al., 2022; Jain et al., 2022)). As for backward policy regularization, we demonstrate that the RBS-GFN exhibits robust performance across a wide range of values for the decay hyperparameter $\beta$, consistently achieving 100% success rate with high sample efficiency, as shown in Fig. 15(b).

### D.4 COMPARISON WITH MODEL-BASED GOAL-CONDITIONED RL

To further demonstrate the effectiveness of our proposed RBS-GFN, we carefully design a sophisticated model-based GC-RL method following MHER (Yang et al., 2021) and Dreamerv3 (Hafner et al., 2023) for additional comparison. Specifically, we consider actor-critic learning introduced in Deamerv3, and employ the model-based imagination technique introduced in MHER to augment the training data. It is noteworthy that while previous model-based methods like MHER augment datasets by forward imagination based on HER, RBS introduces a novel way to sample backward trajectories to increase both data quality and diversity. The results shown in Fig. 16(a) demonstrate that RBS-GFN consistently outperforms this GC-RL method by a large margin.

### D.5 PERFORMANCE ON STOCHASTIC ENVIRONMENTS

To further demonstrate that RBS-GFN can generalize to stochastic dynamics, we build it upon stochastic GFN (Pan et al., 2023d) and consider randomness in the environment following Machado et al. (2018). Specifically, the environment transitions according to the selected action with probability $1 - \alpha$, while with probability $\alpha$ the environment executes a randomly chosen action. Here we take the grid environments for evaluation and set $\alpha = 0.01$ to make the environments stochastic. The

experimental results shown in Fig. 16(d) demonstrate that RBS-GFN can also generalize to stochastic environments and achieve higher success rates compared with the strongest baseline OC-GAFN. Given the inherent randomness in the environments, which can significantly influence goal-reaching strategies, it is reasonable for the overall performance to decline compared to that in deterministic environments.

### D.6    ROBUSTNESS TO DIFFERENT REWARD STRUCTURES

To demonstrate that RBS-GFN is robust to different reward structures, we add additional experiments on the GridWorld tasks, where the agent receives dense rewards (which corresponds to an easier setting compared to the sparse reward case we studied in the main paper). Concretely, the reward is defined as the Manhattan distance between the current state and the desired goal. The results shown in Fig 16(b) demonstrate that RBS-GFN achieves even further performance improvements in the dense rewards setting.

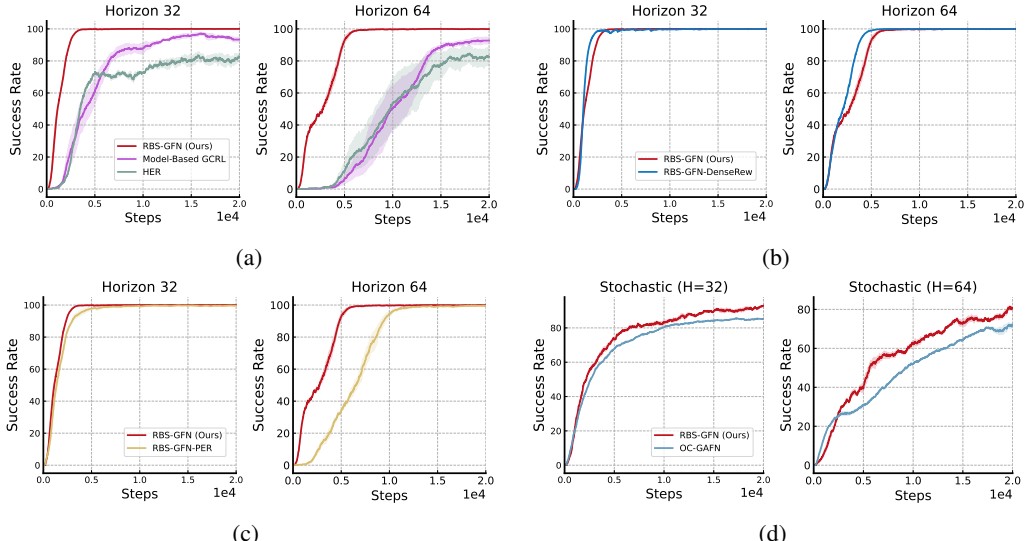

Figure 16: Additional experimental results on the GridWorld benchmark. (a) Success rates compared with an advanced model-based goal-conditioned RL method. (b) RBS-GFN can generalize to the dense rewards structure and gain further performance improvement. (c) RBS-GFN, with the age-based sampling technique, outperforms its variant with PER. (d) Success rates on the stochastic GridWorld environments (Pan et al., 2023d) compared with OC-GAFN (across 3 random seeds). RBS-GFN consistently outperforms the strongest baseline.

### D.7    COMPARISON WITH PRIORITIZED EXPERIENCE REPLAY (PER)

Specifically, we follow the standard PER setting (Schaul et al., 2016) with $\alpha = 0.7$ and $\beta = 0.4$, and we adopt the GC-GFlowNet loss instead of TD error as a priority to suit our scenario. We utilize the open-sourced codes in `https://github.com/Howuhh/prioritized_experience_replay` to implement it. From the results in Fig 16(c), we observe that our age-based sampling outperforms PER by a large margin that learns more efficiently. We hypothesize that this is because the GC-GFlowNet loss is not stable; for some data samples, the loss would remain irreducible and stay high throughout the training process. Consequently, PER restricts training data coverage, leading to reduced overall performance. In contrast, our age-based sampling technique ensures that experiences are leveraged more thoroughly.

### D.8    GENERALIZATION AND VERSATILITY

**Generalization.**    We investigate the generalization ability of our method in more unseen maps. We show the three designed maps that consider different locations of goals and obstacles in Fig. 18(a-c). We also compare our method with baselines in terms of the success rate on these unseen maps. The

experimental results shown in Fig. 18(d-e) demonstrate our proposed RBS method significantly enhances the generalization ability of GC-GFlowNets.

**Versatility**   To demonstrate that our method can also be applied to SubTB (Madan et al., 2023b) learning objective, we provide additional experimental results on the TF Bind sequence generation task. We observe that both RBS-GFN trained with DB (denoted as RBS-GFN(DB) ) and RBS-GFDN trained with SubTB (denoted as RBS-GFN (SubTB)) achieve a $100\%$ success rate. notably, our method RBS gets consistent performance improvement in this task, while RBS-GFN-w/o-RBS almost fails to succeed.

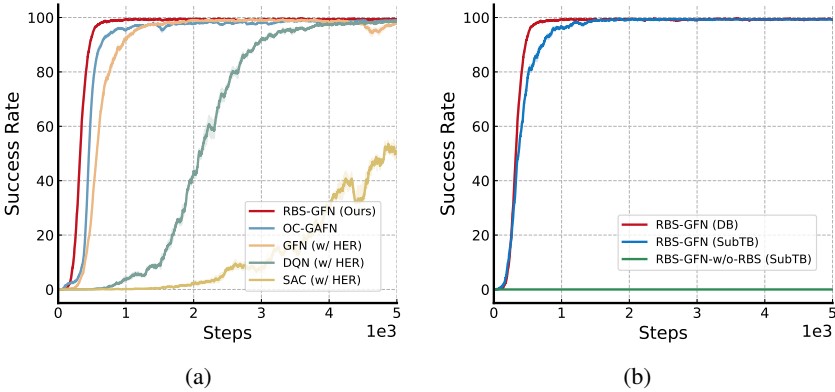

Figure 17: (a) Performance of SAC in TF Bind tasks. (b) Performance of GFN with SubTB in TF Bind tasks.

### D.9   COMPARISON WITH SAC

We investigate the performance of SAC, known as soft actor-critic algorithm (Haarnoja et al., 2018), which is an entropy-regularized RL method rather than standard $\arg\max$ DQN. We evaluate the performance of SAC in the TF Bind sequence generation task. As the action space is this task is discrete, we implement a discrete SAC algorithm based on the codes from CleanRL (Huang et al., 2022). From the results shown in Fig 17(a), we observe that SAC (w/ HER) even performs worse than the DQN algorithm. We hypothesize that it is because SAC prefers new states to maximize the entropy rather than high-reward states to complete the task. Although HER can provide abundant of successful experiences, it is still not enough for SAC to succeed.

## E   LIMITATIONS AND DISCUSSIONS

### E.1   LIMITATIONS AND FUTURE WORK

In this paper, we mainly address key training challenges in GC-GFlowNets problems, and study standard evaluation benchmarks from the GFlowNets literature (Bengio et al., 2021; 2023) with structured tasks (e.g., DNA/RNA generation) where the dynamics are known and well-defined. For some tasks where environment dynamics might be unknown or infeasible to directly model, we can learn a backward dynamic model $f$ to predict the previous state, e.g., $s_{t-1} = f(s_t, a_{t-1})$, following Höftmann et al. (2023); Pan et al. (2023d). With a sufficiently collected dataset, learning a dynamic model is feasible as it can be framed as a regression problem. We hope our work can inspire future research in this promising direction studying unknown dynamics in the environment.

To align with the established and commonly used benchmarks and evaluation protocols in both GFlowNets (Bengio et al., 2023) and GC-GFlowNets (Pan et al., 2023a) literature, our work primarily focuses on deterministic and discrete environments, which have been well-established and studied. While recent theoretical work (Bengio et al., 2023) explore continuous GFlowNets, their practical implementations and applications remain limited, as highlighted in Jain et al. (2023), where training continuous GFlowNets poses significant challenges and scaling them to realistic tasks remains an

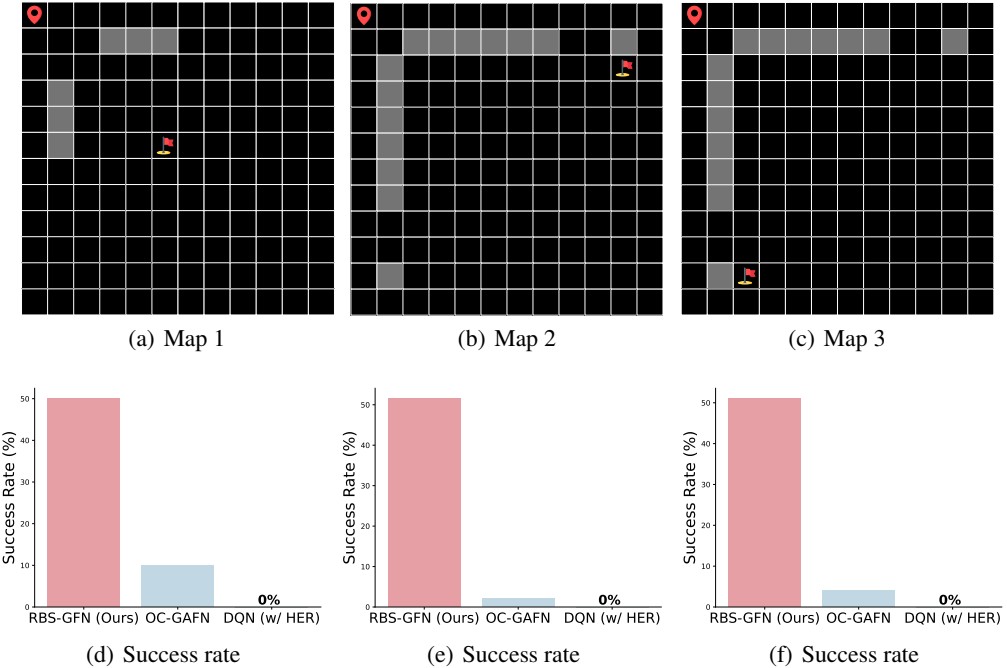

Figure 18: *(a)~(c)*: Additional designed unseen maps to evaluate the generalization ability. *(d)~(e)*: Average success rate over 3 random seeds on these unseen maps for 200 trials.

open problem in the field. We leave the extension of our method to continuous environments for future work.

## E.2 BACKWARD LEARNING IN REINFORCEMENT LEARNING (RL)

Previous works (Goyal et al., 2019; Edwards et al., 2018; Lai et al., 2020; Wang et al., 2021) in model-based RL leverage backward world models to optimize policies for returning to high-value states, while they fail to address the reward sparsity challenges in goal-conditioned RL. Höftmann et al. (2023) use a backward dynamic model to generate trajectories for learning goal-conditioned policies by imitation learning, which sidesteps the requirement of rewards for policy learning. However, its performance heavily relies on the quality of the learned backward model and has only been evaluated in relatively simple maze environments. Model-based goal-conditioned RL has been emerging as a promising direction, which employs a learned world model to imagine future trajectories to improve policy learning in an online (Yang et al., 2021; Charlesworth & Montana, 2020; Wang et al., 2024) or offline manner (Kim et al., 2024; Wang et al., 2023). However, these methods primarily leverage forward imagination through the learned dynamic model to improve policy learning, whereas RBS introduces a novel approach by sampling backward trajectories, thereby enhancing both data quality and diversity.

