# OpenReview forum: "Looking Backward: Retrospective Backward Synthesis for Goal-Conditioned GFlowNets"
_ICLR.cc/2025/Conference — ICLR 2025 Poster_

### Official Review · Reviewer_iEkK · 2024-11-01

**Soundness:** 2
**Presentation:** 3
**Contribution:** 2
**Rating:** 6
**Confidence:** 3

**Summary:**

The paper tackles the challenge of training goal-conditioned Generative Flow Networks (GC-GFlowNets) in environments with sparse rewards and limited offline data. It introduces Retrospective Backward Synthesis (RBS), which synthesizes new backward trajectories to enrich training data, improving sample efficiency and diversity. Experiments demonstrate that RBS significantly improves performance and generalization in various benchmarks.

**Strengths:**

This paper proposes a novel method called Retrospective Backward Synthesis (RBS), which synthesizes new backward trajectories in goal-conditioned GFlowNets to improve the quality and diversity of training trajectories. This approach introduces rich learnable signals, effectively addressing the sparse reward problem.

**Weaknesses:**

1.The experimental tasks are relatively simple and insufficiently comprehensive.

2.The latest goal-conditioned reinforcement learning algorithms are not selected for comparison.

**Questions:**

1.	Age-Based Sampling is a very straightforward technique. How does it compare to previous methods like Prioritized Experience Replay (PER)? Did the authors attempt using PER as well?

2.	Regarding the experimental setup, since you’ve compared your method with reinforcement learning approaches, I assume that these experiments share the same tasks as those in reinforcement learning. If that’s the case, why weren’t newer RL methods selected as baselines? Additionally, for tasks like bit sequence generation, TF binding generation, and AMP generation, is DQN an appropriate baseline?

---

> ### Author Response · Authors · 2024-11-18
> **Response to Reviewer iEkK (Part 1)**
>
> Thank you for your valuable feedback and welcome suggestions for improvements. Here, we carefully address your concerns as follows:
>
> >Q1: The experimental tasks are relatively simple and insufficiently comprehensive.
>
> Thank you for your comment. Our work addresses the fundamental challenges of GC-GFlowNets and studies challenging and standard environments following the literature of GFlowNets [1,2], from illustrative to highly complex scenarios. We begin with the GridWorld environments featuring sparse rewards and long horizons, which serve as clear demonstrations of our method's capabilities across varying horizons. We then advance to high-dimensional, real-world applications such as TF Bind and AMP generation, following established benchmarks in the GC-GFlowNets literature [1]. These tasks involve high dimensionality and significant complexity: the AMP generation task involves $20^{50}$ candidates, while the bit generation task has $2^{100}$ candidates, representing vast state and action spaces that pose significant challenges for goal-conditioned learning. Despite this complexity, our proposed RBS-GFN method consistently achieves nearly 100% success rates on these complex, high-dimensional tasks, demonstrating its robustness and effectiveness in handling challenging environments.
>
> [1] Pan, Ling, Moksh Jain, Kanika Madan, and Yoshua Bengio. "Pre-Training and Fine-Tuning Generative Flow Networks." In The Twelfth International Conference on Learning Representations.
>
> [2] Malkin, Nikolay, Moksh Jain, Emmanuel Bengio, Chen Sun, and Yoshua Bengio. "Trajectory balance: Improved credit assignment in gflownets." Advances in Neural Information Processing Systems 35 (2022): 5955-5967.
>
> >Q2: The latest goal-conditioned reinforcement learning algorithms are not selected for comparison.
>
> Thanks for your suggestion to improve our paper! We compared against goal-conditioned RL baselines considering hindsight experience replay (HER) [3] built upon DQN and SAC (please refer to Appendix C.2 for details). These GC-RL methods are well-suited for our studied tasks, which are structured and discrete. RBS-GFN demonstrates significant improvements over these baselines in both efficiency and performance. Additionally, we remark that GC-RL methods typically yield limited solutions due to their reward-maximizing nature, whereas our approach facilitates more diverse goal-reaching strategies (as demonstrated in our main experimental results).
>
>
> To provide an even more comprehensive comparison, we also include an additional baseline considering a strong model-based GC-RL method following MHER [4] and Dreamerv3 [5], which is based on actor-critic learning as introduced in Deamerv3, and also employs the model-based imagination technique introduced in MHER to augment the training data. It is noteworthy that previous model-based GC-RL methods like MHER augment datasets by forward imagination based on HER, while RBS-GFN introduces a novel way to sample backward trajectories, thereby enhancing both data quality and diversity. RBS-GFN trains GC-FlowNets to learn goal-reaching policies with diverse solutions that improve generalization and robustness. `Please find the details and experimental results in Appendix D.4` The additional experimental results demonstrate that RBS-GFN consistently outperforms this strong GC-RL method by a large margin, further showcasing the superior efficacy of our proposed RBS-GFN method.
>
> [3] Andrychowicz, Marcin, Filip Wolski, Alex Ray, Jonas Schneider, Rachel Fong, Peter Welinder, Bob McGrew, Josh Tobin, OpenAI Pieter Abbeel, and Wojciech Zaremba. "Hindsight experience replay." Advances in neural information processing systems 30 (2017).
>
> [4] Yang, Rui, Meng Fang, Lei Han, Yali Du, Feng Luo, and Xiu Li. "MHER: Model-based Hindsight Experience Replay." In Deep RL Workshop NeurIPS 2021.
>
> [5] Hafner, Danijar, Jurgis Pasukonis, Jimmy Ba, and Timothy Lillicrap. "Mastering diverse domains through world models." arXiv preprint arXiv:2301.04104 (2023).

---

> ### Author Response · Authors · 2024-11-18
> **Response to Reviewer iEkK (Part 2)**
>
> >Q3: Age-Based Sampling is a very straightforward technique. How does it compare to previous methods like Prioritized Experience Replay (PER)? Did the authors attempt using PER as well?
>
> Thank you for your question. We did consider Prioritized Experience Replay (PER) in our initial investigations; however, we observed that it did not perform well on our tasks. As a result, we developed our age-based sampling technique, which operates in a simple yet effective manner to enhance the training efficiency of GC-GFlowNets.
>
> We have also included additional experiments for the performance of PER in Appendix D.7, which demonstrates that it underperforms our proposed age-based sampling techniques. In this additional experiment, we follow standard PER settings [6] and adopt the GC-GFlowNets loss instead of TD error (for GC-RL) as the priority metric to adapt it in our scenario. `Please refer to Appendix D.7 for more details and experimental results` (where age-based sampling outperforms PER by a large margin). We attribute the performance difference induced by PER to the inherent complexity of GC-GFlowNets learning, which differs from standard reinforcement learning. Specifically, the GC-GFlowNets loss exhibits significant instability, and certain data samples may maintain persistently high, irreducible losses throughout the training process [7]. As a result, PER can limit the diversity of the training data in our case since it always assigns high priority to these data samples, leading to reduced overall performance. In contrast, our age-based sampling technique is designed to provide more comprehensive coverage of the training experiences. This approach is simple yet effective and does not require careful hyperparameter tuning, such as the tuning of $\alpha$ and $\beta$ in PER.
>
> [6] Schaul, Tom, John Quan, Ioannis Antonoglou, and David Silver. "Prioritized Experience Replay." In The Sixth International Conference on Learning Representations.
>
> [7] Sujit, Shivakanth, Somjit Nath, Pedro Braga, and Samira Ebrahimi Kahou. "Prioritizing samples in reinforcement learning with reducible loss." Advances in Neural Information Processing Systems 36 (2023): 23237-23258.
>
> >Q4: Regarding the experimental setup, since you’ve compared your method with reinforcement learning approaches, I assume that these experiments share the same tasks as those in reinforcement learning. If that’s the case, why weren’t newer RL methods selected as baselines? Additionally, for tasks like bit sequence generation, TF binding generation, and AMP generation, is DQN an appropriate baseline?
>
> Thanks for your question. Please refer to our response to Q2 for a detailed discussion of our baseline comparisons and additional experiments.
>
> ---
>
> We thank the reviewer for the time and effort in reviewing our work! We would greatly appreciate it if the reviewer could check our responses and the updates in the paper and let us know whether they have adequately addressed your concerns. We are happy to provide further clarification if you have any additional concerns.

---

> ### Author Response · Authors · 2024-11-22
> **Respectful Reminder**
>
> Dear Reviewer,
>
> We hope this message finds you well. Since the discussion phase will end soon, we want to kindly follow up on the response we provided to your valuable comments and questions. Your feedback has been crucial in refining our work, and we greatly appreciate the time and effort you have invested in reviewing our paper. If you have any additional thoughts or questions about our responses, we would be more than happy to address them.
>
> Thank you once again for your insightful review! We look forward to your further feedback.
>
> Best,
>
> The authors

---

> > ### Author Response · Authors · 2024-11-27
> > **Looking forward to your reply!**
> >
> > Dear Reviewer iEkK,
> >
> > As the discussion period will end soon, we'd like to ask if we have addressed your concerns. We believe that we have clarified the concerns and questions raised. Specifically, we included additional experiments to address concerns about comparison with GCRL (`Appendix D.4`) and comparison with PER (`Appendix D.7`). This will be a good improvement to our manuscript. We are happy to provide further clarification if you have any additional concerns. Thanks again for your feedback.
> >
> > Best,
> >
> > The authors

---

> > > ### Author Response · Authors · 2024-11-29
> > > **Follow-up reminder**
> > >
> > > Dear Reviewer iEkK,
> > >
> > > Thank you again for your time and effort in reviewing our work! We would like to bring to your attention that the discussion period will end in about three days, and it has been over ten days since we submitted our rebuttal. We would greatly appreciate it if you could check our responses and the updates in the paper and let us know whether they have adequately addressed your concerns. We believe that we have clarified the concerns and questions raised. We have added additional experiments in `Appendix D.4 and Appendix D.7` for your review.
> > >
> > > Thank you once again for your insightful review! We look forward to your further feedback.
> > >
> > > Best,
> > >
> > > The authors

---

### Official Review · Reviewer_xjJi · 2024-11-02

**Soundness:** 3
**Presentation:** 3
**Contribution:** 3
**Rating:** 8
**Confidence:** 3

**Summary:**

This paper addresses key challenges in goal-conditioned Generative Flow Networks (GFlowNets), specifically the problems of sparse rewards and limited trajectory coverage. The authors introduce Retrospective Backward Synthesis (RBS), a method that generates additional backward trajectories to expand the training data. Their approach aims to improve both the quality and diversity of training trajectories, providing more learning signals in scenarios with sparse rewards. Empirical evaluations demonstrate improved sample efficiency and performance compared to baseline methods across multiple benchmarks.

**Strengths:**

* The paper is well-written and straightforward to understand.
* Retrospective Backward Synthesis (RBS) is introduced with clear motivation, and the paper also presents training techniques such as backward policy regularization.
* Empirical results demonstrate that the proposed method outperforms baselines, showing improved performance and sample efficiency.

**Weaknesses:**

* The evaluation tasks do not include key benchmarks like RNA Generation from Pan et al. (2023a), which limits direct comparison.
* The differences between the proposed RBS method and OC-GAFN are not clearly articulated. A more comprehensive discussion is needed to clarify the specific advantages of the RBS method.
* It remains unclear how goals are defined across the evaluated tasks, which could impact generalizability and reproducibility.

**Questions:**

Please address my concerns in the weakness

---

> ### Author Response · Authors · 2024-11-18
> **Response to Reviewer xjJi**
>
> Thank you for the thorough and detailed review and a positive assessment of our work! Here, we carefully address your concerns as follows:
>
> >Q1: The evaluation tasks do not include key benchmarks like RNA Generation from Pan et al. (2023a), which limits direct comparison.
>
> Thanks for your suggestion to improve our paper! We have included additional experiments in this task for a more thorough comparison. `The experimental results are presented in Appendix D.2`, demonstrating that RBS-GFN consistently outperforms the strongest baseline, OC-GAFN, on the RNA generation task. We also note that RNA generation shares fundamental characteristics with other tasks studied in the paper, e.g., TF Bind and AMP generation.
>
> >Q2: The differences between the proposed RBS method and OC-GAFN are not clearly articulated. A more comprehensive discussion is needed to clarify the specific advantages of the RBS method.
>
> Thanks for your comment. We would like to summarize the differences as follows.
>
> First, our method introduces a novel backward trajectory sampling mechanism. While OC-GAFN relies on Hindsight Experience Replay [1] for trajectory relabelling with failure rewards, which fails to generate novel experiences and requires training two separate GFN models, RBS-GFN trains a single GC-GFN model that generates new trajectories through looking backward. This approach produces high-reward, diverse trajectories that improve both the performance and generalizability of GC-GFlowNets.
>
> We further introduce three key technical innovations to fully realize the effectiveness of RBS-GFN: 1) a backward policy regularization mechanism that is compatible with any GC-GFN learning objective that effectively enhances the diversity of the imagined trajectories; 2) an age-based sampling technique for comprehensive experience utilization during training in a simple yet effective manner; and 3) an intensified reward feedback for improved sample efficiency that strengthens the propagation of learning signal particularly in long-horizon tasks. We also propose a hierarchical-style method to address the high-dimensional and long-horizon AMP generation problems.
>
> Lastly, our method demonstrates significant performance improvements and efficiency on large-scale tasks. By addressing challenging high-dimensional problems based on our proposed new techniques, we successfully handle complex prepend/append MDP cases and achieve a nearly 100% success rate, while previous methods fail completely under limited training budgets.
>
>
> >Q3: It remains unclear how goals are defined across the evaluated tasks, which could impact generalizability and reproducibility.
>
> Thanks for your question. Take the GridWorld environment as an illustrative example. In GC-GFlowNets, we train agents to reach arbitrary goals via diverse goal-reaching solutions, enhancing training efficiency and generalization capabilities. Following the GC-GFlowNets literature [2,3], goals are defined as terminal states in the environment. For example, in an $H\times H$ GridWorld, goals are represented as arbitrary 2D locations $(m,n)$ (where $0\leq m,n < H$). `Moreover, we will open-source all the codes with comprehensive documentation upon acceptance to ensure reproducibility and facilitate future research.`
>
> **Reference**:
>
> [1] Andrychowicz, Marcin, Filip Wolski, Alex Ray, Jonas Schneider, Rachel Fong, Peter Welinder, Bob McGrew, Josh Tobin, OpenAI Pieter Abbeel, and Wojciech Zaremba. "Hindsight experience replay." Advances in neural information processing systems 30 (2017).
>
> [2] Pan, Ling, Moksh Jain, Kanika Madan, and Yoshua Bengio. "Pre-Training and Fine-Tuning Generative Flow Networks." In The Twelfth International Conference on Learning Representations.
>
> [3] Bengio, Yoshua, et al. "Gflownet foundations." The Journal of Machine Learning Research 24.1 (2023): 10006-10060.
>
> ---
>
> We thank the reviewer for the time and effort in reviewing our work! We would greatly appreciate it if the reviewer could check our responses and the updates in the paper and let us know whether they have adequately addressed your concerns. We are happy to provide further clarification if you have any additional concerns.

---

> > ### Author Response · Authors · 2024-11-27
> > **Follow-up**
> >
> > Dear Reviewer xjJi,
> >
> > Thanks again for your positive assessment of our work! Since the discussion phase will end soon, we want to kindly follow up on the response we provided to your valuable comments and questions. Your feedback has been crucial in refining our work, and we greatly appreciate the time and effort you have invested in reviewing our paper. We believe that we have addressed all the raised concerns. If you have any additional thoughts or questions about our responses, we would be more than happy to address them.
> >
> > Best,
> >
> > The authors

---

### Official Review · Reviewer_HyCh · 2024-11-03

**Soundness:** 3
**Presentation:** 4
**Contribution:** 3
**Rating:** 5
**Confidence:** 3

**Summary:**

The paper proposes Retrospective Backward Synthesis (RBS), a novel method to enhance the training of goal-conditioned Generative Flow Networks (GC-GFlowNets). GC-GFlowNets have shown potential in generating diverse sets of high-reward candidates but face challenges due to sparse reward structures and limited coverage of explored trajectories, especially when using offline data. To address these limitations, RBS synthesizes backward trajectories that originate from a desired goal, enriching the training data with high-quality, diverse samples. This approach helps transform unsuccessful action sequences into positive learning experiences, thereby improving sample efficiency and generalizability.

The authors introduce additional techniques, such as reward signal intensification and backward policy regularization, to stabilize training and prevent mode collapse. Empirical results across various benchmarks, including GridWorld and bit sequence generation, demonstrate that RBS outperforms state-of-the-art methods in terms of success rates, sample efficiency, and scalability. Notably, RBS achieves nearly 100% success in large-scale tasks where competing approaches fail, highlighting its robustness and potential for further advancements in GC-GFlowNets.

**Strengths:**

- Backward-Looking Strategy for Enhanced Training: The proposed Retrospective Backward Synthesis (RBS) method utilizes a backward-looking strategy to synthesize trajectories from the goal state, significantly enriching training data. This approach effectively improves sample efficiency by converting failed experiences into successful learning signals, addressing the sparse reward problem.
- Empirical Validation of Sample Efficiency: The paper presents strong empirical results across a range of benchmarks, demonstrating that RBS markedly improves sample efficiency. The method achieves nearly 100% success rates in complex tasks where state-of-the-art baselines fall short, underscoring its practical impact.
- Clear Writing and Presentation: The paper is well-written and presented, with clear explanations, structured methodology, and comprehensive experimental results. The clarity facilitates a strong understanding of both the theoretical and practical aspects of the proposed approach.

**Weaknesses:**

- Scalability and Continuous Environments: The paper’s experiments focus on relatively simple and discrete environments, raising concerns about how well the Retrospective Backward Synthesis (RBS) method would scale to more complex, continuous, real-world tasks. The absence of testing in high-dimensional or continuous state-action spaces limits insights into its broader applicability.
- Tuning Challenges: The method's reliance on hyperparameters, such as reward scaling and backward policy regularization, introduces tuning challenges. While these components are beneficial for stabilizing training, they require careful adjustment, potentially impacting the ease of replication and practical deployment in varied scenarios.
- Lack of Comparison with Model-Based RL: Despite the inherent use of backward trajectory synthesis, which resembles model-based planning, the paper does not compare RBS with established model-based RL approaches such as MBPO or Dreamer. This omission makes it difficult to assess how RBS performs relative to other methods that also utilize environment models for planning and sample efficiency.

**Questions:**

- In algorithm 1) line 6, how do we guarantee that the backward policy could reach $s_0$ from $y$ each time?
- Are tuning for reward intensification and backward policy regularization difficult? What the the effect of hyper-parameters on the performance?

---

> ### Author Response · Authors · 2024-11-18
> **Response to Reviewer HyCh (Part 1)**
>
> Thank you for all the critical comments and valuable suggestions for helping improve the paper. We provide detailed responses to address your concerns below.
> >Q1: Scalability and Continuous Environments: The paper’s experiments focus on relatively simple and discrete environments, raising concerns about how well the Retrospective Backward Synthesis (RBS) method would scale to more complex, continuous, real-world tasks. The absence of testing in high-dimensional or continuous state-action spaces limits insights into its broader applicability.
>
> Thank you for your comment. Our paper focuses on addressing the key challenges of GC-GFlowNets and primarily focuses on high-dimensional discrete environments that are well-established benchmarks in prior GFlowNets research [1,2,3,4]. This includes the challenging and long-horizon GridWorld environments with sparse rewards, which are considered one of the most challenging scenarios due to reward sparsity. More importantly, we tackle practical, high-dimensional, and complex tasks, including TF Bind and AMP generation. For instance, the AMP generation task involves $20^{50}$ possible candidates, while the bit generation task involves $2^{100}$ possible solutions. Despite this enormous and high-dimensional state-action space and complexity that poses significant challenges for GC-GFlowNets, our proposed RBS-GFN method consistently achieves nearly 100% success rates.
>
> Regarding continuous environments, while recent theoretical work [5] explored continuous GFlowNets, their practical implementations and applications remain limited, as highlighted in [6], where training continuous GFlowNets poses significant challenges, and scaling them to realistic tasks remains an open problem in the field. Our focus on discrete spaces aligns with the current state of reliable GFlowNets applications and well-established benchmarks. Nevertheless, the framework of our method is naturally extensible to continuous settings, which also presents an interesting direction in expanding our approach.
>
> [1] Bengio, Yoshua, et al. "Gflownet foundations." The Journal of Machine Learning Research 24.1 (2023): 10006-10060.
>
> [2] Kim, Minsu, Taeyoung Yun, Emmanuel Bengio, Dinghuai Zhang, Yoshua Bengio, Sungsoo Ahn, and Jinkyoo Park. "Local Search GFlowNets." In The Twelfth International Conference on Learning Representations.
>
> [3] Pan, Ling, Moksh Jain, Kanika Madan, and Yoshua Bengio. "Pre-Training and Fine-Tuning Generative Flow Networks." In The Twelfth International Conference on Learning Representations.
>
> [4] Bengio, Emmanuel, Moksh Jain, Maksym Korablyov, Doina Precup, and Yoshua Bengio. "Flow network based generative models for non-iterative diverse candidate generation." Advances in Neural Information Processing Systems 34 (2021): 27381-27394.
>
> [5] Lahlou, Salem, Tristan Deleu, Pablo Lemos, Dinghuai Zhang, Alexandra Volokhova, Alex Hernández-Garcıa, Léna Néhale Ezzine, Yoshua Bengio, and Nikolay Malkin. "A theory of continuous generative flow networks." In International Conference on Machine Learning, pp. 18269-18300. PMLR, 2023.
>
> [6] Jain, Moksh, Tristan Deleu, Jason Hartford, Cheng-Hao Liu, Alex Hernandez-Garcia, and Yoshua Bengio. "Gflownets for ai-driven scientific discovery." Digital Discovery 2, no. 3 (2023): 557-577.
> >Q2: Tuning Challenges: The method's reliance on hyperparameters, such as reward scaling and backward policy regularization, introduces tuning challenges. While these components are beneficial for stabilizing training, they require careful adjustment, potentially impacting the ease of replication and practical deployment in varied scenarios.
>
> Thanks for your comment. Regarding the reward scaling technique, the scaling coefficient $C$ is the only task-dependent hyperparameter that requires tuning, as it depends on the nature of specific tasks and also accommodates different horizons. However, this can be easily done through standard techniques like grid search (similar to tuning conventional hyperparameters such as the learning rate [7,8]). As for backward policy regularization, we demonstrate that the RBS-GFN exhibits robust performance across a wide range of values for the decaying hyperparameter $\beta$, consistently achieving a 100\% success rate with high sample efficiency,  `as summarized in ablation studies in Appendix D.3`. `Moreover, we will open-source all the codes with comprehensive documentation upon acceptance to ensure replication and facilitate future research.`
>
> [7] Malkin, Nikolay, Moksh Jain, Emmanuel Bengio, Chen Sun, and Yoshua Bengio. "Trajectory balance: Improved credit assignment in gflownets." Advances in Neural Information Processing Systems 35 (2022): 5955-5967.
>
> [8] Jain, Moksh, Emmanuel Bengio, Alex Hernandez-Garcia, Jarrid Rector-Brooks, Bonaventure FP Dossou, Chanakya Ajit Ekbote, Jie Fu et al. "Biological sequence design with gflownets." In International Conference on Machine Learning, pp. 9786-9801. PMLR, 2022.

---

> ### Author Response · Authors · 2024-11-18
> **Response to Reviewer HyCh (Part 2)**
>
> >Q3: Lack of Comparison with Model-Based RL: Despite the inherent use of backward trajectory synthesis, which resembles model-based planning, the paper does not compare RBS with established model-based RL approaches such as MBPO or Dreamer. This omission makes it difficult to assess how RBS performs relative to other methods that also utilize environment models for planning and sample efficiency.
>
> Thanks for your suggestion to improve our paper! While MBPO and Dreamerv3 are established model-based RL methods, they are not specifically designed for goal-conditioned learning. To address the concern and to further provide a meaningful comparison with model-based approaches, we follow MHER [9] and Dreamerv3 [10] and carefully designed an additional model-based GC-RL baseline. Specifically, we consider actor-critic learning introduced in Deamerv3, and employ the forward model-based imagination technique introduced in MHER to augment the training data. It is noteworthy that previous model-based GC-RL methods, e.g., MHER, rely on forward imagination with hindsight experience replay, whereas our proposed RBS-GFN method introduces backward trajectory sampling to enhance both data quality and diversity. `Please find the details and experimental results in Appendix D.4`. The additional experimental results demonstrate that RBS-GFN still outperforms the model-based GC-RL method by a large margin and further validate its superior efficiency and effectiveness.
>
> [9] Yang, Rui, Meng Fang, Lei Han, Yali Du, Feng Luo, and Xiu Li. "MHER: Model-based Hindsight Experience Replay." In Deep RL Workshop NeurIPS 2021.
>
> [10] Hafner, Danijar, Jurgis Pasukonis, Jimmy Ba, and Timothy Lillicrap. "Mastering diverse domains through world models." arXiv preprint arXiv:2301.04104 (2023).
>
> >Q4: In algorithm 1) line 6, how do we guarantee that the backward policy could reach $s_0$ from $y$ each time?
>
> In structured environments where GFlowNets typically operate [1,3,4], we can guarantee to reach $s_0$ through well-defined reverse dynamics. Take the GridWorld tasks as an example: given an initial state $s_0$, each action in the action space has a deterministic reverse counterpart (e.g., "up" reverses "down"). The backward sampling process continues until the agent reaches $s_0$, with the reverse action space ensuring reachability through the same paths that exist in the forward direction.
>
> >Q5: Are tuning for reward intensification and backward policy regularization difficult? What is the effect of hyper-parameters on the performance?
>
> Please refer to our response to Q2.
>
> ---
> Thanks to the reviewer for the time and effort in reviewing our work! We would greatly appreciate it if the reviewer could check our responses and the updates in the paper and let us know whether they have adequately addressed your concerns. We hope we have resolved all the concerns, and we would be grateful if you could kindly consider raising the score. We are happy to provide further clarification if you have any additional concerns.

---

> ### Author Response · Authors · 2024-11-22
> **Respectful Reminder**
>
> Dear Reviewer,
>
> We hope this message finds you well. Since the discussion phase will end soon, we want to kindly follow up on the response we provided to your valuable comments and questions. Your feedback has been crucial in refining our work, and we greatly appreciate the time and effort you have invested in reviewing our paper. If our responses have resolved your concerns, could you please consider raising your score? If you have any additional thoughts or questions about our responses, we would be more than happy to address them.
>
> Thank you once again for your insightful review! We look forward to your further feedback.
>
> Best,
>
> The authors

---

> ### Author Response · Authors · 2024-11-27
> **Looking forward to your reply!**
>
> Dear Reviewer HyCh,
>
> As the discussion period will end soon, we'd like to ask if we can provide any more information that could affect your assessment of the paper. We believe that we have clarified the concerns and questions raised. Specifically, we included additional experiments to address concerns about scalability and hyperparameter tuning (`see Appendix D.5 and Appendix D.3`), and we added a model-based GCRL baseline to demonstrate the superior efficacy of our method (`see Appendix D.4`). This will be a good improvement to our manuscript. Thanks again for your feedback.
>
> Best,
>
> The authors

---

> ### Author Response · Authors · 2024-11-29
> **Follow-up reminder**
>
> Dear Reviewer HyCh,
>
> Thank you again for your time and effort in reviewing our work! We would like to bring to your attention that the discussion period will end in about three days, and it has been over ten days since we submitted our rebuttal. We would greatly appreciate it if you could check our responses and the updates in the paper and let us know whether they have adequately addressed your concerns. We believe that we have clarified the concerns and questions raised. We have added additional experiments in `Appendix D.3, Appendix D.4 and Appendix D.5` for your review. We would be grateful if you could kindly consider raising the score if there are no additional concerns.
>
> Thank you once again for your insightful review! We look forward to your further feedback.
>
> Best,
>
> The authors

---

> ### Author Response · Authors · 2024-12-02
> **Reminder**
>
> Dear Reviewer HyCh,
>
> Thank you again for your time and effort in reviewing our work! **We would like to bring to your attention that the discussion period will end today (`less than 24 hours remaining`)**. We would greatly appreciate it if you could check our responses and the updates in the paper and **let us know whether they have adequately addressed your concerns**. We believe that we have clarified the concerns and questions raised. We have added additional experiments in `Appendix D.3, Appendix D.4 and Appendix D.5` for your review. **We would be grateful if you could kindly consider raising the score if there are no additional concerns**.
>
> Thank you once again for your insightful review! We look forward to your further feedback.
>
> Best,
>
> The authors

---

### Official Review · Reviewer_fNHf · 2024-11-05

**Soundness:** 3
**Presentation:** 3
**Contribution:** 3
**Rating:** 8
**Confidence:** 3

**Summary:**

The authors introduces a novel method, Retrospective Backward Synthesis (RBS), aimed at enhancing the training of goal-conditioned Generative Flow Networks (GFlowNets) by synthesizing new backward trajectories. RBS augments "virtual" backward trajectories in
goal-conditioned GFlowNets to enrich training trajectories with enhanced quality and diversity. RBS improves the sample efficiency and performance of GFlowNets across a range of tasks, including sequence generation and biological sequence design.

**Strengths:**

1. The paper identifies and targets a significant issue in the training of goal-conditioned GFlowNets, offering a practical and innovative solution. Augmenting backward trajectories for training is interesting.
2. Comprehensive empirical results are provided, demonstrating the effectiveness of RBS over existing methods on multiple benchmarks.

**Weaknesses:**

1. Limited Discussion on Potential Drawbacks: The paper does not sufficiently address the potential limitations of RBS. For instance, there is no discussion about the computational overhead of synthesizing backward trajectories, nor is there a mention of whether the method is robust to different types of reward structures or environment dynamics.
2. Relevance and Scope of Application: The improvements are made specifically within the context of GC-GFlowNets, which may limit the applicability of the method.
3. Comparison with Diffusion Policies: Given the similarities between the proposed RBS and diffusion policies, a direct comparison would be valuable to understand the unique contributions and differences of RBS.
4. Assumptions on Environment Dynamics: It is unclear whether the proposed RBS method assumes or requires any particular properties of the environment, such as determinism or stochasticity. If the backward dynamics are infeasible or the environment is highly stochastic, the performance of RBS may be affected, and this should be addressed.
5. Quality of Synthetic Trajectories: The paper should include a discussion on how to ensure the quality of the synthesized backward trajectories, especially in cases where such trajectories may not correspond to realistic or feasible paths in the actual environment.
6. Lack of Comparison with Goal-Conditioned RL: Without a comparison to goal-conditioned RL, it is difficult for readers to fully appreciate the relative strengths and weaknesses of GC-GFlowNets. Including such a comparison would provide a more complete picture of the method's positioning within the broader field of goal-directed learning.
7. The authors may further investigate existing literature on augmenting backward trajectories for sample-efficient RL or backward learning in goal-conditioned RL, which makes the paper more comprehensive.

**Questions:**

1. How does RBS compare with diffusion policies, and in what scenarios does RBS offer distinct advantages?
2. Does RBS assume deterministic or stochastic environments, and how does it handle situations where the backward dynamics are not straightforward?
3. How can the authors ensure that the synthesized backward trajectories are meaningful and do not lead to false positives in the learning process?
4. Could the authors include a comparison with goal-conditioned RL methods to highlight the specific benefits of using GC-GFlowNets?

---

> ### Author Response · Authors · 2024-11-18
> **Response to Reviewer fNHf (Part 1)**
>
> Thank you for your thorough comments and valuable feedback! Below, we provide detailed responses to address your concerns.
>
> >Q1: Limited Discussion on Potential Drawbacks: The paper does not sufficiently address the potential limitations of RBS. For instance, there is no discussion about the computational overhead of synthesizing backward trajectories, nor is there a mention of whether the method is robust to different types of reward structures or environment dynamics.
>
> Thanks for your comment. We have added additional experiments and included the discussions in the revision.
>
> (1) Regarding computational efficiency, our RBS-GFN approach is efficient since we only need to synthesize a single $\tau'$ using backward policy $P_B$ from a specified goal, and rollout a minibatch of data at each training step. It is also worth noting that the strongest baseline OC-GAFN requires training two GFN models (i.e., an unconditioned GAFN model and a goal-conditioned GFN model) [1], while our method only needs to train a single goal-conditioned GFN agent, which largely reduces computational requirements in training. To further quantify the computational overhead, we measure the wall-clock time and evaluate performance improvements alongside the corresponding increases in time. `Please refer to the experimental results and details in Figure 15(a) and Appendix D.1.` Our findings show that RBS-GFN achieves a significantly higher success rate in less training time compared to the previous strongest OC-GAFN method.
>
> (2) Regarding the reward structure, as described in Section 2.2, we follow previous goal-conditioned learning settings and evaluate our proposed RBS-GFN approach under the most challenging reward structures: sparse and episodic rewards [1] (which also reflects real-world applications), where the agent receives a binary reward only when upon reaching the terminal state. Nevertheless, our method can also generalize well to different reward structures, such as dense rewards. We conduct additional experiments in the GridWorld benchmark, where the agent receives dense rewards (which corresponds to an easier setting compared to the sparse reward case we studied), and the results demonstrate that RBS-GFN achieves even further performance improvements. `Please find the detailed experimental results in Figure 16(b) and Appendix D.6.`
>
> (3) Concerning environment dynamics, we have conducted extensive experiments across different types of environments [1] to demonstrate the robustness of our method, including GridWorld with different layouts with varying unseen obstacles shown in Figure 6 and Figure 7 and also varying horizons in Figure 3, and TF Bind and AMP generation tasks with different vocabularies and lengths. We highlight that the proposed RBS-GFN approach consistently achieves the best performance on a variety of challenging environments. Since current GFlowNets literature primarily focuses on these deterministic tasks, we study these established settings following standard evaluation protocol as in [1]. To further demonstrate that RBS-GFN can generalize to stochastic dynamics, we build it upon Stochastic GFN [2, 3] and consider randomness in the environment following [4]. `The experimental results shown in Figure 16(d) and Appendix D.5 ` demonstrate that RBS-GFN can also generalize to stochastic environments and achieve higher success rates compared with the strongest baseline OC-GAFN. We hope our work can provide advancements in training GC-GFlowNets and inspire further research to systematically explore this promising direction.
>
> [1] Pan, Ling, Moksh Jain, Kanika Madan, and Yoshua Bengio. "Pre-Training and Fine-Tuning Generative Flow Networks." In The Twelfth International Conference on Learning Representations.
>
> [2] Pan, Ling, Dinghuai Zhang, Moksh Jain, Longbo Huang, and Yoshua Bengio. "Stochastic generative flow networks." In Uncertainty in Artificial Intelligence, pp. 1628-1638. PMLR, 2023.
>
> [3] Bengio, Yoshua, Salem Lahlou, Tristan Deleu, Edward J. Hu, Mo Tiwari, and Emmanuel Bengio. "Gflownet foundations." The Journal of Machine Learning Research 24, no. 1 (2023): 10006-10060.
>
> [4] Machado, Marlos C., Marc G. Bellemare, Erik Talvitie, Joel Veness, Matthew Hausknecht, and Michael Bowling. "Revisiting the arcade learning environment: Evaluation protocols and open problems for general agents." Journal of Artificial Intelligence Research 61 (2018): 523-562.

---

> ### Author Response · Authors · 2024-11-18
> **Response to Reviewer fNHf (Part 2)**
>
> >Q2: Relevance and Scope of Application: The improvements are made specifically within the context of GC-GFlowNets, which may limit the applicability of the method.
>
> Thanks for your comment. GC-GFlowNets represent a significant advancement over traditional GFlowNets, as traditional GFlowNets require training from scratch for each new task, while GC-GFlowNets offer a more efficient paradigm that enables quick adaptation to downstream tasks through pre-training a GFN agent to reach arbitrary goals, and GC-FlowNets is the key in this paradigm. As demonstrated in [1], a well-trained GC-GFlowNet agent can efficiently adapt to various downstream tasks (even without policy re-training). In this paper, we propose a novel method, RBS-GFN, to enhance the performance of GC-GFlowNets by addressing the key challenges of extremely sparse rewards and limited coverage of trajectories.  Our RBS-GFN method enhances this capability by improving the efficiency and quality of pre-trained goal-reaching strategies, as validated in Section 5.5 where RBS-GFN offers substantial improvements during this adaptation process. The practical impact of our method is also significant when considering the broad applications of GFlowNets, as previous works [5, 6, 7, 8] have demonstrated their utility in critical domains, including biological sequence design, molecule generation, combinatorial optimization, and language models. By improving the fundamental goal-reaching capabilities of GC-GFlowNets, our method has the potential to advance applications across these domains. Additionally, while our paper primarily addresses challenges within GC-GFlowNets, we believe that our insights underlying backward synthesis in this paper (e.g., diverse, efficient, and robust goal-reaching) are also beneficial for addressing key challenges faced in the field of goal-conditioned learning.
>
> [5] Jain, Moksh, Emmanuel Bengio, Alex Hernandez-Garcia, Jarrid Rector-Brooks, Bonaventure FP Dossou, Chanakya Ajit Ekbote, Jie Fu et al. "Biological sequence design with gflownets." In International Conference on Machine Learning, pp. 9786-9801. PMLR, 2022.
>
> [6] Bengio, Emmanuel, Moksh Jain, Maksym Korablyov, Doina Precup, and Yoshua Bengio. "Flow network based generative models for non-iterative diverse candidate generation." Advances in Neural Information Processing Systems 34 (2021): 27381-27394.
>
> [7] Zhang, David W., Corrado Rainone, Markus Peschl, and Roberto Bondesan. "Robust Scheduling with GFlowNets." In The Eleventh International Conference on Learning Representations.
>
> [8] Hu, Edward J., Moksh Jain, Eric Elmoznino, Younesse Kaddar, Guillaume Lajoie, Yoshua Bengio, and Nikolay Malkin. "Amortizing intractable inference in large language models." In The Twelfth International Conference on Learning Representations.
> >Q3: Comparison with Diffusion Policies: Given the similarities between the proposed RBS and diffusion policies, a direct comparison would be valuable to understand the unique contributions and differences of RBS.
>
> Thank you for your comment. We would like to clarify that Diffusion Policies [9] utilize a diffusion and denoising strategy, similar to diffusion models, to model actions, which is applied in offline learning to enhance the generalization ability and expressiveness of policies. In contrast, RBS is designed to improve the performance of GC-GFlowNets by synthesizing high-quality and diverse backward trajectories. RBS effectively addresses the key challenges of reward sparsity and limited trajectory coverage in GC-GFlowNets learning, which helps the agent learn more effective goal-reaching strategies in an efficient manner, which is even more helpful when given offline data, and we will include the discussion about diffusion policies in the paper. These two methods are different as they employ distinct approaches to solve different problems. If the reviewer refers to a different diffusion policy paper related to GC-GFlowNets, we will be happy to provide additional discussion and a direct comparison.
>
> [9] Chi, Cheng, Zhenjia Xu, Siyuan Feng, Eric Cousineau, Yilun Du, Benjamin Burchfiel, Russ Tedrake, and Shuran Song. "Diffusion policy: Visuomotor policy learning via action diffusion." The International Journal of Robotics Research (2023): 02783649241273668.

---

> ### Author Response · Authors · 2024-11-18
> **Response to Reviewer fNHf (Part 3)**
>
> >Q4: Assumptions on Environment Dynamics: It is unclear whether the proposed RBS method assumes or requires any particular properties of the environment, such as determinism or stochasticity. If the backward dynamics are infeasible or the environment is highly stochastic, the performance of RBS may be affected, and this should be addressed.
>
> Thanks for your comment. Following the established and commonly used benchmarks and evaluation protocols in both GFlowNets [3] and GC-GFlowNets [1] literature, our work primarily focuses on deterministic environments, and the extensions of GFlowNets to stochastic environments [2, 10] have not been well-studied. To address this concern, we have conducted additional experiments, detailed in Appendix D.5 following the setup in Stochastic GFN [2], where the results demonstrate that RBS-GFN can be extended to stochastic environments.  (Please refer to our response to Q1 for details).
>
> [10] Zhang, Dinghuai, Ling Pan, Ricky TQ Chen, Aaron Courville, and Yoshua Bengio. "Distributional GFlowNets with Quantile Flows." Transactions on Machine Learning Research.
>
> >Q5: Quality of Synthetic Trajectories: The paper should include a discussion on how to ensure the quality of the synthesized backward trajectories, especially in cases where such trajectories may not correspond to realistic or feasible paths in the actual environment.
>
> Thanks for your comment. We synthesize a backward trajectory $\tau'$ starting from a goal $y$. When employing $\tau'$ for training, we reverse it to guarantee $\tau'$ starts from the initial state $s_0$, maintaining consistency with the forward trajectory $\tau$. The feasibility and dynamics-consistency of our synthesized paths are naturally preserved as GFlowNets primarily deal with structured tasks (e.g., DNA/RNA generation) where the dynamics are known and well-defined [1, 3, 4]. As a result, the synthesized backward trajectory $\tau'$ guarantees that it reaches the desired goal $y$ and receives a positive successful reward of $1$. For scenarios where environment dynamics might be unknown or infeasible to model directly, we have added a discussion in Appendix E.1 and provide potential approaches to address the problem, such as learning a transition dynamics model like [3]. We hope our work can inspire future research into the exploration of unknown environmental dynamics.
>
> >Q6: Lack of Comparison with Goal-Conditioned RL: Without a comparison to goal-conditioned RL, it is difficult for readers to fully appreciate the relative strengths and weaknesses of GC-GFlowNets. Including such a comparison would provide a more complete picture of the method's positioning within the broader field of goal-directed learning.
>
> Thanks for your suggestion to improve our paper! We would like to clarify that we have included comparisons with goal-conditioned RL baselines considering hindsight experience replay (HER) [11] built upon DQN and SAC (please refer to Appendix C.2 for details) in our paper. RBS-GFN demonstrates significant improvements over these baselines in both efficiency and performance. Additionally, we remark that GC-RL methods typically yield limited solutions due to their reward-maximizing nature, whereas our approach facilitates more diverse goal-reaching strategies (as demonstrated in our main experimental results), which is beneficial for generalization and robustness.
>
> To further demonstrate the effectiveness of our proposed RBS-GFN approach for addressing the concern, we conduct additional comparisons with a sophisticated model-based GC-RL method following Model-based Hindsight Experience Replay (MHER) [12] and Dreamerv3 [13]. Specifically, we consider actor-critic learning introduced in Deamerv3, and employ the model-based imagination technique introduced in MHER to augment the training data. It is noteworthy that while previous model-based methods like MHER rely on forward imagination based on HER [11], RBS introduces a novel way to sample backward trajectories, thereby enhancing both data quality and diversity. `Please find the details and experimental results in Appendix D.4.` The results demonstrate that RBS-GFN consistently outperforms this carefully designed GC-RL method by a large margin.
>
> [11] Andrychowicz, Marcin, Filip Wolski, Alex Ray, Jonas Schneider, Rachel Fong, Peter Welinder, Bob McGrew, Josh Tobin, OpenAI Pieter Abbeel, and Wojciech Zaremba. "Hindsight experience replay." Advances in neural information processing systems 30 (2017).
>
> [12] Yang, Rui, Meng Fang, Lei Han, Yali Du, Feng Luo, and Xiu Li. "MHER: Model-based Hindsight Experience Replay." In Deep RL Workshop NeurIPS 2021.
>
> [13] Hafner, Danijar, Jurgis Pasukonis, Jimmy Ba, and Timothy Lillicrap. "Mastering diverse domains through world models." arXiv preprint arXiv:2301.04104 (2023).

---

> ### Author Response · Authors · 2024-11-18
> **Response to Reviewer fNHf (Part 4)**
>
> >Q7: The authors may further investigate existing literature on augmenting backward trajectories for sample-efficient RL or backward learning in goal-conditioned RL, which makes the paper more comprehensive.
>
> Thanks for your comment. We have included additional discussions in Appendix E.2 in the revision.
>
> >Q8: How does RBS compare with diffusion policies, and in what scenarios does RBS offer distinct advantages?
>
> Please refer to our detailed response to Q3.
>
> >Q9: Does RBS assume deterministic or stochastic environments, and how does it handle situations where the backward dynamics are not straightforward?
>
> Please refer to our responses to Q1, Q4, and Q5.
>
> >Q10: How can the authors ensure that the synthesized backward trajectories are meaningful and do not lead to false positives in the learning process?
>
> Please refer to our responses to Q5.
>
> >Q11: Could the authors include a comparison with goal-conditioned RL methods to highlight the specific benefits of using GC-GFlowNets?
>
> Please refer to our responses to Q6.
>
> ---
> Thanks to the reviewer for the time and effort in reviewing our work! We would greatly appreciate it if the reviewer could check our responses and the updates in the paper and let us know whether they have adequately addressed your concerns. We are happy to provide further clarification if you have any additional concerns.

---

> ### Author Response · Authors · 2024-11-22
> **Respectful Reminder**
>
> Dear Reviewer,
>
> We hope this message finds you well. Since the discussion phase will end soon, we want to kindly follow up on the response we provided to your valuable comments and questions. Your feedback has been crucial in refining our work, and we greatly appreciate the time and effort you have invested in reviewing our paper. If you have any additional thoughts or questions about our responses, we would be more than happy to address them.
>
> Thank you once again for your insightful review! We look forward to your further feedback.
>
> Best,
>
> The authors

---

> > ### Author Response · Authors · 2024-11-27
> > **Looking forward to your reply!**
> >
> > Dear Reviewer fNHf,
> >
> > As the discussion period will end soon, we'd like to ask if we have addressed your concerns. We believe that we have clarified the concerns and questions raised. Specifically, we included additional experiments to address concerns about computation overhead (`Appendix D.1`), generalizing to different environment dynamics and reward structures (`Appendix D.5 and D.6`), and comparison with GCRL (`Appendix D.4`). This will be a good improvement to our manuscript.  We are happy to provide further clarification if you have any additional concerns. Thanks again for your feedback.
> >
> > Best,
> >
> > The authors

---

> ### Comment · Reviewer_fNHf · 2024-11-27
>
> Thanks for the detailed response, which has addressed many of my concerns. The current version is better. I increase my score to 8. Good luck.

---

> > ### Author Response · Authors · 2024-11-27
> > **Thank you for increasing your score!**
> >
> > Thank you for increasing your score! We sincerely appreciate your thoughtful and detailed comments for improving the quality of our paper. We are grateful for your recognition of our work and honored by your positive evaluation!

---

### Author Response · Authors · 2024-11-18
**General Response**

We thank all of the reviewers for their time and insightful comments. Furthermore, we are very glad to find that reviewers generally recognize the effectiveness, novelty and significance of our work:
- **Method**: The paper identifies and targets a significant issue in the training of goal-conditioned GFlowNets, offering a practical and innovative solution [**fNHf**]. This approach effectively improves sample efficiency by converting failed experiences into successful learning signal [**HyCh**]. RBS is introduced with clear motivation [**xjJi**]. This approach introduces rich learnable signals, effectively addressing the sparse reward problem [**HyCh, iEkK**]. This paper proposes a novel method called RBS [**iEkK**].

- **Presentation**: The paper is well-written and presented, with clear explanations, structured methodology, and comprehensive experimental results [**HyCh**]. The paper is well-written and straightforward to understand [**xjJi**].

- **Experiments**: Comprehensive empirical results are provided [**fNHf**]. The paper presents strong empirical results across a range of benchmarks, demonstrating that RBS markedly improves sample efficiency. Empirical results demonstrate that the proposed method outperforms baselines across a range of benchmarks, showing improved performance and sample efficiency [**HyCh, xjJi**].

Meanwhile, we thank all the reviewers for their helpful and constructive feedback to improve the quality of our work again. We have carefully updated our paper to incorporate the valuable suggestions from the reviewers. We summarize the revisions and added experiments as follows:

- [**fNHf**] We add additional experiments to demonstrate the computation efficiency of our method in Appendix D.1. Results are illustrated in Figure 15(a).
- [**fNHf**] We demonstrate that RBS-GFN can gain further performance improvements in a different reward structure, i.e., dense rewards. Please see the results in Figure 16(b) and Appendix D.6.
- [**fNHf, HyCh**] We evaluate RBS-GFN on stochastic environments to demonstrate its generalizability in Appendix D.5. Results are shown in Figure 16(d).
- [**fNHf, HyCh, iEkK**] We compare our method, i.e., RBS-GFN, with an advanced model-based goal-conditoned RL method. RBS-GFN consistently outperforms this method significantly. Details can be found in Figure 16(a) and Appendix D.4.
- [**HyCh**] We add an additional ablation study to show that RBS-GFN is robust to our introduced decay hyperparameter $\beta$. We provide the results in Figure 15(b) and Appendix D.3.
- [**xjJi**] We further demonstrate the superior efficacy of RBS-GFN on the RNA generation benchmark. Results are given in Figure 15\(c\) and Appendix D.2.
- [**iEkK**] We compare our proposed age-based sampling technique with PER. Results shown in Figure 16(c) and Appendix D.7 demonstrate that our method is more effective.
---
All changes in the new pdf are highlighted in red.

We hope to have addressed all the raised concerns and would be happy to respond to further questions and suggestions.

---

### Meta-Review · Area_Chair_4XNJ · 2024-12-20

**Metareview:**

This paper introduces Retrospective Backward Synthesis (RBS), a novel method that addresses challenges in training goal-conditioned Generative Flow Networks (GC-GFlowNets) by enriching training data with synthesized high-quality, diverse backward trajectories. Additional techniques are introduced, such as reward signal intensification and backward policy regularization, to prevent mode collapse. Empirical results across various benchmarks demonstrating its superior performance in discrete environments.

The reviewers generally acknowledge the paper's contribution in its potential for advancing GC-GFlowNet applications, considering it both novel and solid. The paper is well-written, with a clear motivation, structured methodology, and comprehensive experimental results. Empirical evaluations demonstrate that RBS effectively addresses the sparse reward problem and significantly outperforms state-of-the-art methods in terms of sample efficiency, success rates, and scalability.

Despite these strengths, some weaknesses and limitations are noticeable in the original manuscript. First, the experiments are limited to discrete environments, raising concerns about the scalability of RBS to continuous real-world tasks. The evaluation tasks also omit some key benchmarks, such as RNA generation. Comparisons with alternative approaches, such as diffusion policies, goal-conditioned RL, and model-based RL methods, are absent, leaving the unique advantages and positioning of RBS unclear. Additionally, the paper does not adequately address the potential computational overhead of synthesizing backward trajectories or its robustness to different reward structures and environment dynamics. Furthermore, questions about the quality of synthesized trajectories, the feasibility of backward dynamics in stochastic environments, and the tuning challenges associated with hyperparameters like reward intensification remain unanswered.

The authors’ responses offer detailed clarifications to these questions and provide additional experimental results. Notably, they address the inherent difficulty of the benchmarks used in the paper, compare their approach with goal-conditioned RL, present experiments on RNA generation, demonstrate generalization to different environments and reward structures, and explain hyperparameter tuning along with its robustness.

Three out of four reviewers recommend acceptance. While the remaining reviewer did not provide follow-up feedback on the authors' response, it seems that the concerns raised have been addressed. Additionally, although two positive reviewers did not respond to the authors’ rebuttal, their initial assessments were favorable. I support the reviewers' recommendation and suggest that the authors carefully address the main points raised in the reviews, particularly focusing on additional analyses and experiments, when preparing the final version.

**Additional Comments On Reviewer Discussion:**

Three out of four reviewers recommend acceptance. While the remaining reviewer did not provide follow-up feedback on the authors' response, it seems that the concerns raised have been addressed. Additionally, although two positive reviewers did not respond to the authors’ rebuttal, their initial assessments were favorable. I support the reviewers' recommendation and suggest that the authors carefully address the main points raised in the reviews, particularly focusing on additional analyses and experiments, when preparing the final version.

---

### Decision · Program_Chairs · 2025-01-22

Accept (Poster)